# A Temporal Difference Method
# for Stochastic Continuous Dynamics

**Haruki Settai**    **Naoya Takeishi**    **Takehisa Yairi**

The University of Tokyo
{sharuki,ntake,yairi}@g.ecc.u-tokyo.ac.jp

## Abstract

For continuous systems modeled by dynamical equations such as ODEs and SDEs, Bellman's principle of optimality takes the form of the Hamilton-Jacobi-Bellman (HJB) equation, which provides the theoretical target of reinforcement learning (RL). Although recent advances in RL successfully leverage this formulation, the existing methods typically assume the underlying dynamics are known a priori because they need explicit access to the drift and diffusion coefficients to update the value function following the HJB equation. We address this inherent limitation of HJB-based RL; we propose a model-free approach still targeting the HJB equation and the corresponding temporal difference method. We prove exponential stability of the induced continuous-time dynamics, and we empirically demonstrate the resulting advantages over transition–kernel–based formulations. The proposed formulation paves the way toward bridging stochastic control and model-free reinforcement learning.

## 1   Introduction

Reinforcement learning (RL) has been successfully applied in various domains, ranging from discrete systems like board games (Silver et al., 2018) to systems that are continuous in both state and time such as robotic control (Kober et al., 2013). RL research has been advancing through a variety of approaches, and much of the work has focused on improving methods by refining objective functions (Schulman et al., 2015, 2017), balancing exploration and exploitation (Haarnoja et al., 2018), and developing more effective architectures (Hafner et al., 2019). These efforts have significantly advanced the field, leading to the development of various successful algorithms.

In contrast to the studies primarily targeting algorithmic components or optimization techniques, we focus on the continuity of time and explore what we call continuous RL, where the dynamics of systems are described by ordinary differential equations (ODEs) or stochastic differential equations (SDEs), which is a relatively underexplored aspect of RL. Although many RL methods have been applied, sometimes heuristically, to both discrete and continuous systems, the continuity of the system has not necessarily been fully exploited, even when it is known in advance. Laying a foundation for utilizing prior knowledge of continuity is important toward more effective learning and decision-making methods, particularly for practical applications such as robot control and autonomous driving, which typically fall into the target of continuous RL.

A way to incorporate prior knowledge of the system's continuity into the learning process is to use the Hamilton-Jacobi-Bellman (HJB) equation. In the Bellman equation, continuity is encoded in the transition kernel. However, in model-free RL, where transitions are approximated using samples, this continuity information is lost because the transition kernel is not explicitly modeled. On the other hand, the HJB equation retains the continuity information in the argument of the expectation rather than in the transition kernel. This property allows the HJB equation to keep taking continuity into

39th Conference on Neural Information Processing Systems (NeurIPS 2025).

account even under sample-based approximations. However, because the HJB equation depends on the coefficient functions of the system's dynamics, prior work has largely been limited to model-based approaches (Munos and Bourgine, 1997; Yıldız et al., 2021).

In this paper, we introduce a temporal difference (TD) method based on the HJB equation, namely *differential TD* (dTD), achieved through sample-based approximation of the expectation term in the HJB equation. dTD enables policy evaluation without requiring knowledge or estimation of the system dynamics, while incorporating the continuity of the dynamics into the learning process. It is compatible with on-policy methods such as A2C (Mnih et al., 2016) and PPO (Schulman et al., 2017), and we demonstrate its effectiveness on Mujoco (Todorov et al., 2012) tasks including Hopper, HalfCheetah, Ant, and Humanoid. The codes for the proposed method are available at `https://github.com/4thhia/differential_TD`.

## 2 Related Work

**Deterministic Dynamics**  The study of continuous RL for ODE systems can be traced back to studies such as Baird (1994); Munos (1997); Doya (2000); Munos (2006). Baird (1994) discovered that the Q-function collapses in continuous RL, which was rigorously proven and extended to deep RL in Tallec et al. (2019). In Munos (1997), model-free approaches for ODE systems were first studied. Doya (2000) was the first to introduce TD for ODE systems and extended it to TD($\lambda$) and actor-critic. Munos (2006) investigated the estimation of policy gradients for ODE systems and proposed a pathwise derivative approach. More recently, Vamvoudakis and Crofton (2017) developed a model-free RL framework for deterministic linear systems. Kim et al. (2021) proposed a model-free Q-learning approach in which the control is derived from the HJB equation, while the learning target is based on the conventional Bellman equation. Yıldız et al. (2021) introduced a model-based method that leverages the Neural ODE framework to enable continuous-time optimization using learned system dynamics.

**Stochastic Dynamics**  One of the earliest works on RL in SDE systems is Munos and Bourgine (1997), which takes a model-based approach. However, research in this direction remained largely unexplored until recently. In the past few years, a growing body of work has emerged that aims to establish theoretical foundations for RL in stochastic dynamics. Wang et al. (2020) introduced an entropy-regularized relaxed control formulation and provided a comprehensive analysis in the LQR setting. Tang et al. (2022) further demonstrated the well-posedness of the HJB equation within this relaxed control framework. Jia and Zhou (2022b) showed that Bellman optimality is equivalent to maintaining the martingale property of a suitably defined stochastic process, and proposed a corresponding algorithm. Building on this approach, Jia and Zhou (2022a, 2023) proposed actor-critic and Q-learning algorithms for finite-horizon SDE systems, respectively. Zhao et al. (2020) extended key theoretical tools such as the state visitation distribution and the performance difference lemma to the continuous-time setting and applied them to TRPO and PPO. Despite these advances, most of the above approaches (i) operate in a model-based regime and, in addition, (ii) are restricted to finite-horizon settings that require access to full trajectory information and/or (iii) assume a linear dynamics model. In contrast, our work proposes a simple model-free TD method that is applicable to general nonlinear stochastic dynamics without requiring knowledge of the SDE coefficients. We demonstrate its effectiveness on standard continuous-control RL benchmarks, going beyond prior work that has been mostly confined to toy SDE examples.

## 3 Background

### 3.1 Problem Setting

We consider a continuous-time RL setting where the state space is $\mathcal{S} \subset \mathbb{R}^n$ and the action space is $\mathcal{A}$. We model the state dynamics by the following controlled SDE:

$$dS_t = \mu(S_t, A_t)dt + \sigma(S_t, A_t)dB_t, \tag{1}$$

where $\mu : \mathcal{S} \times \mathcal{A} \to \mathbb{R}^n$, $\sigma : \mathcal{S} \times \mathcal{A} \to \mathbb{R}^{n \times m}$, and $(B_t)_{t \geq 0}$ is the $m$-dimensional Brownian motion. Note that the state evolution is influenced by both the inherent noise in the system as well as the randomness induced by the stochastic policy $\pi : \mathcal{S} \to \mathcal{P}(\mathcal{A})$, where $\mathcal{P}(\mathcal{A})$ is the space of probability

distribution over the action space. Thus, the expectation related to this SDE is expressed as $\mathbb{E}_{p_\pi}[\cdot]$, where $p_\pi$ denotes the transition kernel induced by (1). For simplicity, we assume that the stochastic process (1) is well-defined; see Appendix A.1 for a detailed justification. We here focus on SDE systems because we can recover results for ODE systems in the limit of $\sigma = 0$.

We primarily focus on the model-free setting, where the agent has no prior knowledge of the dynamics coefficients $\mu$ and $\sigma$ in Eq. (1). Throughout this paper, the term RL refers to this model-free setting unless specifically stated otherwise (e.g., as model-based RL).

## 3.2 Continuous RL

The dynamics governed by the SDE in Eq. (1) exhibit the Markov property (informally, the infinitesimal evolution depends on the past only through $(S_t, A_t)$), and thus continuous RL falls within the general framework of an MDP. For example, the Bellman optimality equation can be written as follows:

$$V^*(s_t) = \max_\pi \mathbb{E}_{p_\pi} \left[ \int_t^{t'} e^{-\gamma(\tau - t)} \rho(S_\tau, A_\tau) d\tau + e^{-\gamma(t' - t)} V^*(S_{t'}) \middle| S_t = s_t \right],$$

where $\rho : \mathcal{S} \times \mathcal{A} \to \mathbb{R}$ is the reward rate function, and $\gamma \in (0, \infty)$ is the constant discount rate. Here, $V^*(s)$ is the optimal value function, defined as:

$$V^*(s) := \max_\pi \mathbb{E}_{p_\pi} \left[ \int_t^\infty e^{-\gamma(\tau - t)} \rho(S_\tau, A_\tau) d\tau \middle| S_t = s \right].$$

When learning based on discrete observations of a continuous system, such as in simulations, one can discretize the problem as follows:

$$V^*(s_t) = \max_\pi \mathbb{E}_{p_\pi} \left[ \rho(s_t, A_t) \Delta t + e^{-\gamma \Delta t} V^*(S_{t + \Delta t}) \right], \tag{2}$$

where $\Delta t$ is a small time interval and need not be constant. This shows that standard RL methods can be applied to continuous RL, although the use of Q-functions in continuous time requires care (Tallec et al., 2019). However, this discretization-based approach can be inefficient in the model-free setting, where the expectation is approximated using samples. In particular, the system dynamics enter only through the transition kernel, and approximating the expectation by samples does not explicitly exploit the underlying continuity of the dynamics. As a result, the agent may fail to leverage structural information that is specific to continuous-time systems.

## 3.3 Natural Target of Continuous RL

We focus on TD learning, a fundamental approach in RL. TD learning naturally targets two types of Bellman equations: the Bellman optimality equation and the Bellman expectation equation. Combining these with the choice of value function (V or Q) yields four candidate targets. Among them, the V-Bellman optimality equation is immediately excluded as it is not compatible with model-free framework. Moreover, as we will show in Section 4, the Q-Bellman optimality equation is challenging to learn effectively with TD methods in continuous RL due to the maximization operator. Thus, in continuous RL, policy evaluation is the most natural scope.

This leaves two remaining candidates: the V- and Q-Bellman expectation equations. While the Q-Bellman expectation equation is compatible with our methodology, Q-functions often degenerate in continuous RL (Tallec et al., 2019), introducing unnecessary complexity. We therefore focus on policy evaluation via the V-Bellman expectation equation,

$$V^\pi(s_t) = \mathbb{E}_{p_\pi} \left[ \rho(s_t, A_t) \Delta t + e^{-\gamma \Delta t} V^\pi(S_{t + \Delta t}) \right]. \tag{3}$$

## 3.4 Contrast with Stochastic Control

Much of the prior work on continuou RL originates from the field of stochastic control. These approaches often start by directly introducing the HJB equation, which is the continuous counterpart of the Bellman optimality equation (2). For instance, the HJB equation can be expressed as: (see

Appendix A.2 for more detail):

$$V^*(s_t) = \frac{1}{\gamma} \max_\pi \; \mathbb{E}_\pi \Bigg[ \rho(s_t, A_t) + \sum_{i=1}^n \mu^i(s_t, A_t) \frac{\partial V^*(s)}{\partial s^i} \bigg|_{s=s_t}$$
$$+ \frac{1}{2} \sum_{i=1}^n \sum_{j=1}^n [\sigma(s_t, A_t)\sigma^\top(s_t, A_t)]^{ij} \frac{\partial^2 V^*(s)}{\partial s^i \partial s^j} \bigg|_{s=s_t} \Bigg], \tag{4}$$

where $\mu^i$ and $[\sigma\sigma^\top]^{ij}$ denote the $i$-th element of $\mu$ and the $(i,j)$-th element of $\sigma\sigma^\top$, respectively.

However, introducing the HJB equation in continuous RL typically restricts the methodology to model-based approaches, since the update requires explicit access to $\mu$ and $\sigma$ (and the same issue persists even when using Q-functions; see Section 4). Thus, for TD learning, the HJB equation offers no clear advantage over the standard Bellman formulation (2).

In contrast, our goal is to design model-free algorithms that explicitly leverage the continuity of the underlying dynamics, which naturally leads us to introduce a variant of the HJB equation built upon the V-Bellman expectation equation (3).

## 4  TD Method for Stochastic Continuous Dynamics

A natural way to inform the agent that the system follows an SDE is to embed the model directly into the update rule. As discussed in Section 3.2, with the standard Bellman expectation equation (3) this information is lost when we pass to a sample-based approximation, which drops the expectation subscript $p_\pi$ where the model is encoded. Therefore, by encoding the SDE information in the argument rather than in the subscript of the expectation, one can expect to keep this information from being lost even under sample approximation. This can be achieved by further transforming the Bellman expectation equation using the SDE, i.e., by expanding $V(S_{t+\Delta t})$ via Itô's formula, resulting in a variant of the HJB equation (the derivation is identical to that for the HJB equation; see Appendix A.2 for more details):

$$V^\pi(s_t) = \frac{1}{\gamma} \mathbb{E}_\pi \Bigg[ \rho(s_t, A_t) + \sum_{i=1}^n \mu^i(s_t, A_t) \frac{\partial V^\pi(s)}{\partial s^i} \bigg|_{s_t} + \frac{1}{2} \sum_{i=1}^n \sum_{j=1}^n [\sigma(s_t, A_t)\sigma^\top(s_t, A_t)]^{ij} \frac{\partial^2 V^\pi(s)}{\partial s^i \partial s^j} \bigg|_{s_t} \Bigg]. \tag{5}$$

We call this equation the HJB for a fixed policy.

Now, are we all ready to implement model-free value iteration just by approximating the expectation on the right-hand side of (5) with samples? The answer is no because the argument of the expectation includes the coefficient functions of the SDE, $\mu$ and $\sigma$, making it impossible to directly approximate the expectation with samples.

### 4.1  Deriving TD from the HJB equation

We now present our main theoretical result. It gives the foundation for our model-free formulation of temporal-difference learning based on the HJB equation. The idea is that the drift and diffusion terms in the HJB equation can be equivalently expressed using limits of sample-based finite differences.

**Proposition 1.** *When a stochastic process $(S_t)_{t\geq 0}$ follows the SDE in* (1)*, we have*

$$\mathbb{E}_{p_\pi} \left[ \mu^i(s_t, A_t) \right] = \lim_{\Delta t \to 0} \mathbb{E}_{p_\pi} \left[ \frac{S^i_{t+\Delta t} - s^i_t}{\Delta t} \right] \tag{6}$$

*and*

$$\mathbb{E}_{p_\pi} \left[ [\sigma(s_t, A_t)\sigma^\top(s_t, A_t)]^{ij} \right] = \lim_{\Delta t \to 0} \mathbb{E}_{p_\pi} \left[ \frac{(S^i_{t+\Delta t} - s^i_t)(S^j_{t+\Delta t} - s^j_t)}{\Delta t} \right]. \tag{7}$$

*Proof.* For the first claim, we expand the $i$-th component of the SDE (1) using the Ito formula:

$$S^i_{t+\Delta t} = s^i_t + \mu^i(s_t, A_t)\Delta t + \sum_{j=1}^m \sigma^{ij}(s_t, A_t)(B^j_{t+\Delta t} - B^j_t) + O(\Delta t^{\frac{3}{2}}). \tag{8}$$

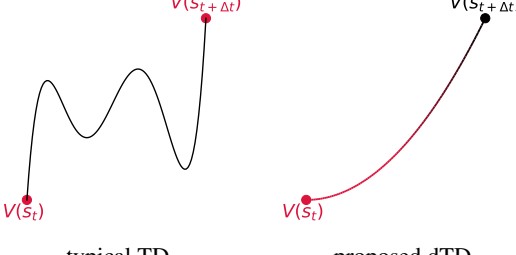

typical TD        proposed dTD

Figure 1: Qualitative difference between the typical TD method and the proposed dTD method; the objects in red indicate what is adjusted by each temporal difference. (*Left*) In the typical TD method, the values of $\hat{V}$ are adjusted to minimize the TD error. (*Right*) In the dTD method, the gradient and the second derivative of $\hat{V}$ at $s_t$ are adjusted to minimize the dTD error.

Since $B_{t+\Delta t}^j - B_t^j$ follows a zero-mean Gaussian and is independent of the state and the action,

$$\mathbb{E}_{p_\pi}\left[\sum_{j=1}^{m}\sigma^{ij}(s_t, A_t)(B_{t+\Delta t}^j - B_t^j)\right] = \sum_{j=1}^{m}\mathbb{E}_\pi\left[\sigma^{ij}(s_t, A_t)\right]\mathbb{E}\left[B_{t+\Delta t}^j - B_t^j\right] = 0.$$

By taking the expectation of both sides of (8) and then letting $\Delta t \to 0$, the terms in $O(\Delta t^{\frac{3}{2}})$ vanish, and we obtain (6). For the second part of the claim, we begin by considering the product:

$$(S_{t+\Delta t}^i - s_t^i)(S_{t+\Delta t}^j - s_t^j)$$

$$= \mu^i(s_t, A_t)\mu^j(s_t, A_t)\Delta t^2 + \sum_{k=1}^{m}\sum_{l=1}^{m}\sigma^{ik}(s_t, A_t)\sigma^{jl}(s_t, A_t)(B_{t+\Delta t}^k - B_t^k)(B_{t+\Delta t}^l - B_t^l)$$

$$+ \mu^i(s_t, A_t)\Delta t\sum_{k=1}^{m}\sigma^{jk}(s_t, A_t)(B_{t+\Delta t}^k - B_t^k) + \mu^j(s_t, A_t)\Delta t\sum_{k=1}^{m}\sigma^{ik}(s_t, A_t)(B_{t+\Delta t}^k - B_t^k) + O(\Delta t^{\frac{3}{2}}).$$

$$(9)$$

and then take the expectation of both sides. Using the fact

$$\mathbb{E}\left[(B_{t+\Delta t}^k - B_t^k)(B_{t+\Delta t}^l - B_t^l)\right] = \delta_{kl}\Delta t,$$

where $\delta_{kl} = 1$ if $k = l$ and $\delta_{kl} = 0$ otherwise, we can take the expectation of both sides of equation (9) and then let $\Delta t \to 0$, which yields (7). $\qquad\square$

**Remark 1.** *Note that since $S_{t+\Delta t}$ in equations (5) and (6) is sampled under the policy $\pi$, our method is not applicable to off-policy settings such as value iteration or Q-learning. This limitation reflects a fundamental distinction: our formulation relies on the HJB equation under a fixed policy (i.e., policy evaluation), rather than the classical HJB equation involving maximization over all policies.*

From Proposition 1, the HJB equation (5) can be reformulated as

$$V^\pi(s_t) = \frac{1}{\gamma}\lim_{\Delta t \to 0}\mathbb{E}_{p_\pi}\left[\rho(s_t, A_t) + \sum_{i=1}^{n}\frac{S_{t+\Delta t}^i - s_t^i}{\Delta t}\frac{\partial V^\pi(s)}{\partial s^i}\bigg|_{s_t}\right.$$
$$\left. + \frac{1}{2}\sum_{i=1}^{n}\sum_{j=1}^{n}\frac{(S_{t+\Delta t}^i - s_t^i)(S_{t+\Delta t}^j - s_t^j)}{\Delta t}\frac{\partial^2 V^\pi(s)}{\partial s^i \partial s^j}\bigg|_{s_t}\right].$$

As we have rearranged the HJB equation so that the argument of expectation does not depend on the model, $\mu$ and $\sigma$, we can construct a temporal-difference update directly from it. We refer to this update as *differential temporal difference (dTD)* and expect it to be particularly effective when the observation interval $\Delta t$ is small.

**Definition 1** (differential temporal difference). *Let $\Delta t > 0$ be a time step and $\widehat{V}$ denote an estimated value function. The dTD is defined as:*

$$\text{dTD} := \frac{1}{\gamma}\left(\rho(s_t, a_t) + \sum_{i=1}^{n}\frac{s_{t+\Delta t}^i - s_t^i}{\Delta t}\frac{\partial\widehat{V}(s)}{\partial s^i}\bigg|_{s_t}\right.$$
$$\left. + \frac{1}{2}\sum_{i=1}^{n}\sum_{j=1}^{n}\frac{(s_{t+\Delta t}^i - s_t^i)(s_{t+\Delta t}^j - s_t^j)}{\Delta t}\frac{\partial^2\widehat{V}(s)}{\partial s^i \partial s^j}\bigg|_{s_t}\right) - \widehat{V}(s_t).$$

$$(10)$$

As illustrated in Figure 1, unlike conventional TD methods based on transition kernels, dTD encourages the learning of a smooth value function by incorporating the continuity of the state space, even under sample-based approximation. We note that the version of dTD for ODE systems can be recovered by simply removing the term corresponding to the diffusion coefficient $\sigma$.

## 4.2 Convergence Analysis

A standard TD convergence analysis relies on the Bellman operator being a contraction. In contrast, the fixed-policy HJB operator involves unbounded differential operators and is not a contraction in general, making the usual arguments inapplicable. Moreover, a parameter-space analysis is technically challenging since the update depends on state derivatives of the value function. Therefore, we adopt an idealized function-space analysis and study the continuous-time limit of the induced dynamics using PDE techniques. For the convergence analysis, we rewrite the second-order term in matrix form for notational convenience.

We first define the key operators for the fixed-policy HJB equation.

**Definition 2** (Fixed-policy HJB operator). *Given a stationary Markov policy $\pi$, discount rate $\gamma > 0$, and functions $\mu, \sigma, \rho$, define the policy-averaged coefficients*

$$\bar{\mu}(s) := \mathbb{E}_{A \sim \pi(\cdot|s)}[\mu(s, A)], \qquad D(s) := \mathbb{E}_{A \sim \pi(\cdot|s)}[\sigma(s, A)\sigma^\top(s, A)], \qquad \bar{\rho}(s) := \mathbb{E}_{A \sim \pi(\cdot|s)}[\rho(s, A)].$$

*The fixed-policy HJB operator $T$ maps $V : S \to \mathbb{R}$ to*

$$(TV)(s) := \frac{1}{\gamma}\Big(\bar{\rho}(s) + (\mathcal{L}^\pi V)(s)\Big),$$

*where $\mathcal{L}^\pi$ is the infinitesimal generator defined below.*

**Definition 3** (Infinitesimal generator). *The infinitesimal generator $\mathcal{L}^\pi$ maps any $C^2$ function $V$ to*

$$(\mathcal{L}^\pi V)(s) := \bar{\mu}(s) \cdot \nabla V(s) + \frac{1}{2}\text{tr}\big(D(s)\nabla^2 V(s)\big).$$

We analyze the iterative scheme

$$V_{k+1} = V_k + \eta_k(TV_k - V_k)$$

through its continuous-time limit

$$\frac{\partial V(t)}{\partial t} = TV(t) - V(t),$$

which corresponds to an idealized setting where the function $V$ can be updated directly in function space. Convergence is characterized by asymptotic stability of the unique fixed point satisfying $TV = V$. Since we consider a fixed policy (and thus no maximization operator), this fixed-point equation reduces to a linear elliptic PDE. Defining the expected reward rate $\bar{\rho}(s) := \mathbb{E}_{\pi(\cdot|s)}[\rho(s, A)]$, the equation $V = TV$ is equivalent to

$$(\gamma I - \mathcal{L}^\pi)V(s) = \bar{\rho}(s).$$

We then apply standard elliptic PDE theory (via the Lax–Milgram theorem) to the bilinear form associated with $(\gamma I - \mathcal{L}^\pi)$, which guarantees existence, uniqueness, and stability of the solution, and hence convergence of the induced dynamics. We leave a full analysis under function approximation to future work.

**Assumption 1.** *We assume:*

1. *(**Domain**) $S \subset \mathbb{R}^n$ is bounded with $C^2$ boundary $\partial S$.*

2. *(**Coefficients**) $\bar{\rho} \in L^2(S)$. The diffusion $D \in W^{1,\infty}(S; \mathbb{R}^{n \times n})$ is symmetric, and the effective drift $b := \bar{\mu} - \frac{1}{2}\text{div } D$ belongs to $L^\infty(S; \mathbb{R}^n)$.*

3. *(**Uniform ellipticity**) There exists $\alpha > 0$ such that $\xi^\top D(s)\xi \geq \alpha\|\xi\|^2$ for all $s \in S$, $\xi \in \mathbb{R}^n$.*

4. *(**Reflecting boundary**) The Neumann condition $n \cdot D\nabla V = 0$ holds on $\partial S$.*

5. *(**Coercivity / discount**) $\gamma > \|b\|_\infty^2/\alpha$.*

We then apply standard elliptic PDE theory (via the Lax–Milgram theorem) to the bilinear form associated with $(\gamma I - \mathcal{L}^\pi)$ to guarantee existence and uniqueness of the fixed point. The exponential stability of the induced continuous-time dynamics is established separately below.

**Lemma 1** (Existence and Uniqueness of the Fixed Point). *Under Assumption 1, the linear elliptic PDE*

$$(\gamma I - \mathcal{L}^\pi)V(s) = \bar{\rho}(s) \quad for\ s \in S$$

*admits a unique weak solution $V^\pi \in H^1(S)$.*

**Proposition 2** (Exponential Stability of the Dynamics). *Let $V^\pi \in H^1(S)$ be the unique weak solution of $(\gamma I - \mathcal{L}^\pi)V = \bar{\rho}$. Under Assumption 1, the solution $V(t)$ of*

$$\frac{\partial V(t)}{\partial t} = TV(t) - V(t)$$

*satisfies, for any $V(0) \in H^1(S)$,*

$$\|V(t) - V^\pi\|_{L^2(S)} \leq \exp(-\lambda t)\, \|V(0) - V^\pi\|_{L^2(S)},$$

*where one may take*

$$\lambda = \frac{1}{\gamma}\left(\gamma - \frac{\|b\|_\infty^2}{\alpha}\right) > 0, \qquad b := \bar{\mu} - \tfrac{1}{2}\mathrm{div}\, D.$$

# 5   Method

This section outlines our method for applying dTD in deep reinforcement learning. Because dTD relies on function approximation, we restrict our attention to the deep RL regime, representing the value function with a neural network. A concise pseudocode listing is provided in Appendix B.1; here we explain the loss formulation and the $\beta$-dTD stabilization strategy.

## 5.1   Loss function

In deep RL, TD methods typically use a fixed target, known as the TD target, $r(s,t) + \gamma_{\text{discrete}} V(s_{t+1})$, as the teacher and aim to approximate the prediction $V(s_t)$ by minimizing the squared error between them. Although it may seem natural, by analogy with classical TD, to treat the $V(s_t)$ term in the dTD (10) as the prediction and regard the remaining terms as the dTD target, this is in fact unnecessary. As shown in Appendix A.2, the terms that appear in (5) and thus in (10) are derived through a series of transformations, and thus in dTD we no longer interpret the terms other than $V(s_t)$ as a low-variance estimate of the Bellman error. Consequently, the split between prediction and target is a design choice.

We examine two different ways of defining the prediction and target from the rhs of (10).

- As a baseline, we first consider a naive formulation following the typical TD-style decomposition, that is, we treat $V(s_t)$ as the prediction. We refer to this approach as **naive-dTD**.

- On the other hand, since TD methods are intended to learn the value function $V$ rather than its derivatives, we treat the value-based terms as the target and regard the derivative-based terms as the prediction. We term such a parametrization simply as **dTD** hereafter.

These two variants are summarized in Table 1. As introduced in the next section, we empirically found that dTD performed significantly better than naive-dTD.

## 5.2   Hybrid scheme for stabilizing dTD

Although Proposition 2 establishes a convergence analysis, it relies on an idealized setting and does not cover the practical instability that can arise from function approximation errors in deep RL. Consequently, we cannot a priori guarantee that plain dTD operates stably and efficiently in practice.

To make the critic update more robust, we linearly combine the classical TD error with the dTD error, using weights $1 - \beta$ and $\beta$, respectively; we call the resulting update $\beta$-dTD. The TD part supplies the empirical stability that underpins most deep RL algorithms, whereas the dTD part injects gradient information from the continuous dynamics, accelerating learning when the underlying assumptions are approximately satisfied. We hypothesize that $\beta$-dTD can strike a balance to stabilize learning progress and potentially improve convergence behavior in practice.

Table 1: Comparison of target and prediction terms in TD methods. Here, $\Delta s_t^i := s_{t+\Delta t}^i - s_t^i$ denotes the $i$-th component of the state transition over a small time interval $\Delta t$.

| | Target | Prediction |
|---|---|---|
| TD | $r(s,t) + \gamma_{\text{discrete}} V(s_{t+1})$ | $V(s_t)$ |
| naive-dTD | $\rho(s_t, a_t) + \sum_{i=1}^{n} \dfrac{\Delta s_t^i}{\Delta t} \dfrac{\partial V(s)}{\partial s^i}\Big\vert_{s_t}$ $+\dfrac{1}{2}\sum_{i=1}^{n}\sum_{j=1}^{n} \dfrac{\Delta s_t^i \Delta s_t^j}{\Delta t} \dfrac{\partial^2 V(s)}{\partial s^i \partial s^j}\Big\vert_{s_t}$ | $\gamma V(s_t)$ |
| dTD | $-\rho(s_t, a_t) + \gamma V(s_{t+\Delta t})$ | $\sum_{i=1}^{n} \dfrac{\Delta s_t^i}{\Delta t} \dfrac{\partial V(s)}{\partial s^i}\Big\vert_{s_t}$ $+\dfrac{1}{2}\sum_{i=1}^{n}\sum_{j=1}^{n} \dfrac{\Delta s_t^i \Delta s_t^j}{\Delta t} \dfrac{\partial^2 V(s)}{\partial s^i \partial s^j}\Big\vert_{s_t}$ |

# 6 Experiments

## 6.1 Modification for discrete environment compatibility

In our theoretical framework, we work with continuous rewards (i.e., reward rate function) and a specific form of the discount ratio $e^{-\gamma}$, which is not directly compatible with the discrete discount ratio $\gamma_{\text{discrete}}$. To address this, we adjusted the reward and discount ratio following the same approach discussed in Tallec et al. (2019). The continuous reward formulation can be approximated by:

$$\int_0^\infty e^{-\gamma t} \rho(s_t, a_t) dt \approx \sum_{k=0}^{\infty} e^{(-\gamma \Delta t)k} \rho(s_{k\Delta t}, a_{k\Delta t}) \Delta t.$$

In this approximation, $\rho(s_t, a_t)\Delta t$ corresponds to the discrete reward $r$, and $e^{-\gamma \Delta t}$ corresponds to the discrete discount ratio $\gamma_{\text{discrete}}$. Thus, we can establish the following relationship:

$$\rho(s_t, a_t) = \frac{r(s_t, a_t)}{\Delta t} \quad \text{and} \quad \gamma = -\frac{1}{\Delta t} \log(\gamma_{\text{discrete}}).$$

This adjustment ensures that the observed discrete rewards are properly scaled to align with the continuous reward formulation used in the dTD method. With this scaling, dTD can be computed as

dTD Target : $-r(s_t, a_t) - \log(\gamma_{\text{discrete}})V(s_{t+\Delta t})$ and

dTD Prediction : $\sum_{i=1}^{n}(s_{t+\Delta t}^i - s_t^i)\dfrac{\partial V(s)}{\partial s_i}\Big\vert_{s_t} + \dfrac{1}{2}\sum_{i=1}^{n}\sum_{j=1}^{n}(s_{t+\Delta t}^i - s_t^i)(s_{t+\Delta t}^j - s_t^j)\dfrac{\partial^2 V(s)}{\partial s_i \partial s_j}\Big\vert_{s_t}.$

## 6.2 Experiment design

**Environment** We conducted experiments with the Brax[1] library (Freeman et al., 2021) in the following environments: Hopper, HalfCheetah, Ant and Humanoid. Each environment provides a mid- to high-dimensional state space, with the number of state components varying across environments: Hopper (11 dimensions), HalfCheetah (17 dimensions), Ant (27 dimensions) and Humanoid (244 dimensions). In each environment, at every step, we perturbed each state component by adding noise in the form of

$$s_i \leftarrow s_i + \text{coef} \times |s_i| \times \text{noise},$$

where noise $\sim \mathcal{N}(0, 1)$, and tested for three values of coef $= 0.00, 0.01, 0.05$. By adding this process noise, we aim to simulate with SDE systems, with the case of coef $= 0.00$ representing the limit case corresponding to ODEs. The specific time-step values used for each environment, which are not directly used in the learning process but are important to ensure they are small enough, are: Hopper: $\Delta t = 0.008$, HalfCheetah: $\Delta t = 0.05$, Ant: $\Delta t = 0.05$ and Humanoid: $\Delta t = 0.015$.

---

[1] https://github.com/google/brax

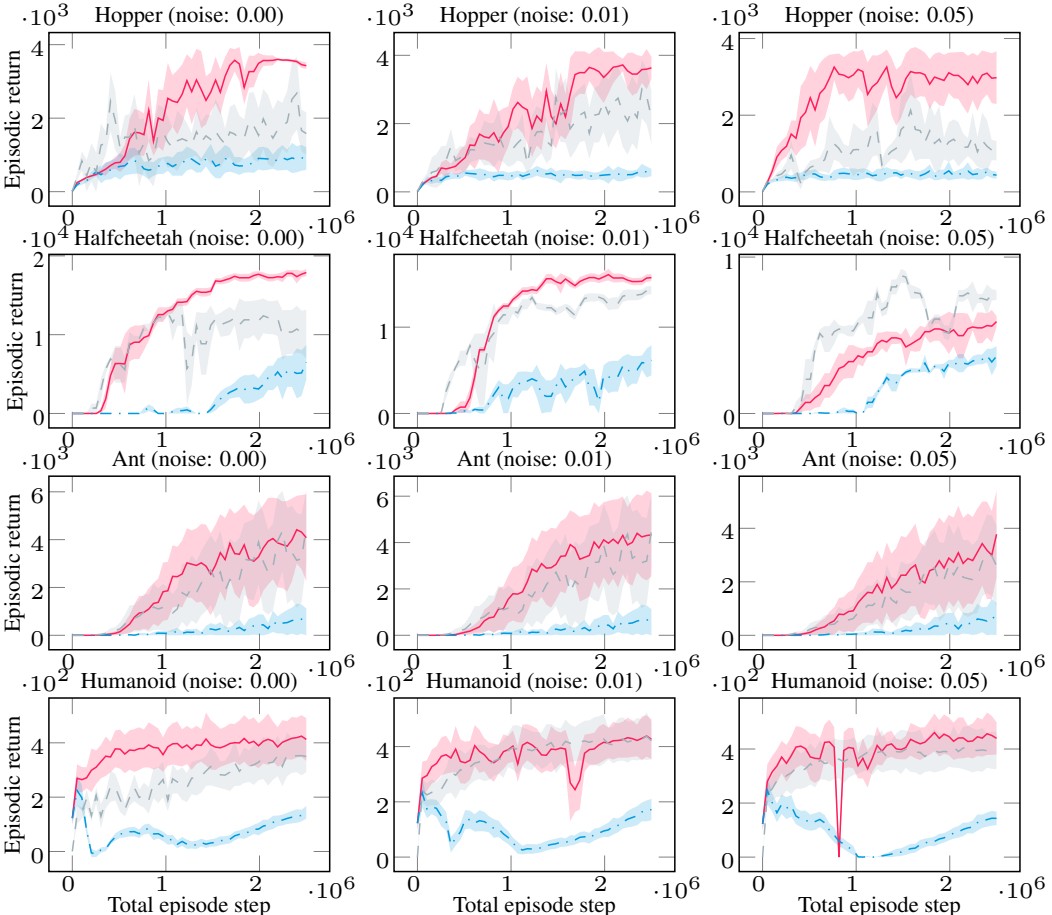

Figure 2: Performance of TD, $\beta$-naive-dTD, and $\beta$-dTD on continuous control benchmark. These results were obtained using the PPO algorithm. Each column corresponds to different noise levels (coef = 0.00, 0.01, 0.05), and each row corresponds to different environments. The tuned $\beta$ values for $\beta$-naive-dTD were 0.08, 0.07, 0.23, 0.02 and for $\beta$-dTD were 0.57, 0.74, 0.24, 0.33 in Hopper, HalfCheetah, Ant, and Humanoid, respectively.

**Baseline**   Various methods have been developed specifically for settings such as ODE (Tallec et al., 2019), LQR (Vamvoudakis and Crofton, 2017), or time-dependent Q functions with finite horizons (Jia and Zhou, 2023), but are often incompatible with the current deep RL framework (Kim et al., 2021) or rely on model-based assumptions (Munos and Bourgine, 1997), making them unsuitable for comparison with our proposed method. We chose to use standard TD methods as baselines and experimented with TD, $\beta$-naive-dTD, and $\beta$-dTD, using A2C(Mnih et al., 2016) and PPO (Schulman et al., 2017). In this paper, we present the results using PPO as the primary focus, and the results using A2C are provided in the Appendix B.4 for supplementary details.

**Hyperparameter tuning**   For hyperparameter tuning, we applied the DEHB (Awad et al., 2021), a multi-fidelity method that is currently considered the most effective method in RL (Eimer et al., 2023). While we performed hyperparameter tuning for the standard PPO algorithm as well, we also reference the official tuning results from Freeman et al. (2021) for fair comparison. Additional details about the hyperparameter search space can be found in Appendix B.

## 6.3   Results and discussion

**Comparative evaluation**   We compare (the variants of) the proposed method and the baseline:

**($\beta$-naive-dTD vs. $\beta$-dTD)** In Figure 3, we can observe that $\beta$-dTD consistently outperforms $\beta$-naive-dTD. In all the environments, the optimized values of $\beta$ for $\beta$-naive-dTD were quite small, suggesting that the effective update rule of $\beta$-naive-dTD became close to that of the standard TD. Despite such a fact, however, the performance of $\beta$-naive-dTD remains significantly worse than the standard TD. There are two possible explanations: (1) the $\beta$ value chosen for $\beta$-naive-dTD was actually still not small enough to fully eliminate the adverse effect of the naive-dTD term; and (2) the TD-related parameters in $\beta$-naive-dTD were only suboptimally tuned because hyperparameter tuning resources were allocated mainly to optimizing $\beta$. These factors may jointly account for the unexpectedly poor performance of $\beta$-naive-dTD.

**(TD vs. $\beta$-dTD)** As shown in Figure 3, $\beta$-dTD outperforms TD or achieves comparable performance in all cases. While the degree of improvement varies, the final performance of TD and $\beta$-dTD tends to converge, which is not very surprising because both dTD and TD are derived from the same Bellman equation, and the resulting value functions should thus be similar to each other eventually. Nevertheless, dTD has the advantage of implicitly utilizing continuity information during training, which enables it to make more informative updates. Consequently, although the final performance may be comparable, $\beta$-dTD tends to show a faster rate of improvement relative to TD.

**Significance of dTD** In contrast to $\beta$-naive-dTD, the weight $\beta$ in $\beta$-dTD is not exceedingly small. Notably, in the Halfcheetah environment, $\beta$ assumes a relatively large value of 0.74. This indicates that dTD retains a meaningful impact on the learning process.

**Impact of process noise** In terms of robustness to process noise, both $\beta$-dTD and TD exhibit similar performance. When the noise level is coef $= 0.01$, neither method experiences significant degradation in performance. However, when the noise level is increased to coef $= 0.05$, both $\beta$-dTD and TD show similar reduction in performance, particularly in environments like Ant and Halfcheetah.

## 7 Conclusion

We have presented differential TD (dTD), a temporal difference method based on the HJB equation. In contrast to approaches based on transition kernels, the proposed method can incorporate the continuity of dynamics into the learning process without knowing the dynamics. We have shown empirical results for a variety of continuous control environments with different time intervals. The empirical results highlight the potential advantages of dTD in terms of learning speed and efficiency while also implying that stability concerns may exist in practice, which led to the introduction of the robust $\beta$-dTD update. Although the current paper focuses on the theoretical development of dTD, these observations are useful and also warrant further empirical exploration.

We have also analyzed the conditions under which the continuous-time dynamics of the HJB equation exhibit exponential stability toward the unique fixed point. This stability property, proven using techniques from linear elliptic PDE theory, is crucial for showing the theoretical convergence of the idealized iterative scheme. However, a drawback is that the sufficient conditions we identify, such as the requirement for a bounded domain and uniform ellipticity (Assumption 1), are often hard to maintain or verify in the context of deep RL with function approximation.

Future work includes bridging the gap between the theoretical exponential stability and the practical stability of dTD updates (e.g., by ensuring the coercivity condition in practice), reducing the variance of learning by improved estimators or regularization, and extending the wide range of existing TD-based techniques to the dTD framework.

## Acknowledgements

NT was supported by JST PRESTO JPMJPR24T6, JSPS JP20K19869, and JSPS JP25H01454.

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

# A    Mathematical Details

## A.1    Justification for the Continuous RL Formulation

In Section 3, we modeled the evolution of the state under a stochastic policy $\pi$ by the controlled SDE

$$dS_t = \mu(S_t, A_t)\, dt + \sigma(S_t, A_t)\, dB_t, \quad A_t \sim \pi(\cdot|S_t).$$

Here, the control is applied in the form of action samples drawn from a stochastic policy at each time step. While this formulation closely reflects the sampling-based behavior in RL, it raises a technical challenge: the presence of external randomness in addition to the intrinsic Brownian noise introduces analytical difficulties. As a result, the well-definedness of this SDE is not immediately obvious.

To address this issue, many prior works (e.g., Wang et al. (2020); Jia and Zhou (2022b,a, 2023); Zhao et al. (2020)) adopt the averaged dynamics, denoted by $(\widetilde{S}_t)_{t\geq 0}$, whose distribution at each time $t$ is known to coincide with that of the original one under the same initial condition (Wang et al., 2020). Specifically, the averaged dynamics is defined as

$$d\widetilde{S}_t = \widetilde{\mu}(\widetilde{S}_t, \pi)dt + \widetilde{\sigma}(\widetilde{S}_t, \pi)d\widetilde{B}_t,$$

where $\widetilde{\mu}(s, \pi) = \int_{\mathcal{A}} \mu(s, a)\pi(a)da$, $\widetilde{\sigma}(s, a) = \left(\int_{\mathcal{A}} \sigma(s, a)\sigma^\top(s, a)\pi(a)da\right)^{\frac{1}{2}}$ and $(\widetilde{B}_t)_{t\geq 0}$ is the $m$-dimensional Brownian motion. Since the averaged dynamics no longer involves the external randomness induced by stochastic action selection, its well-definedness is ensured by classical SDE theory under standard assumptions such as Lipschitz continuity and a linear growth condition.

Since the marginal distributions of the two dynamics coincide, the corresponding value functions also coincide:

$$
\begin{aligned}
V^\pi(s) &= \mathbb{E}_{p_\pi}\left[\int_t^\infty e^{-\gamma(\tau-t)}\rho(S_\tau, A_\tau)\, d\tau \,\middle|\, S_t = s\right] \\
&= \mathbb{E}_{\widetilde{p}}\left[\int_t^\infty e^{-\gamma(\tau-t)}\widetilde{\rho}(\widetilde{S}_\tau, \pi)\, d\tau \,\middle|\, \widetilde{S}_t = s\right] \\
&=: \widetilde{V}^\pi(s),
\end{aligned}
$$

where $\widetilde{\rho}(s, \pi) := \int_{\mathcal{A}} \rho(s, a)\,\pi(a)\, da$. Hence the value function above is itself well defined and raises no analytical issues.

## A.2    Ito formula

The Bellman equation is given by:

$$V^*(s_t) = \max_\pi \mathbb{E}_{p_\pi}\left[\rho(s_t, A_t)\Delta t + e^{-\gamma\Delta t}V^*(S_{t+\Delta t})\right].$$

Assuming that a stochastic process $(S_t)_{t\geq 0}$ follows the SDE (1), the term $V^*(S_{t+\Delta t})$ can be further expanded using Itô's lemma:

$$V^*(s_t) = \max_\pi \; \mathbb{E}_{p_\pi}\left[\rho(s_t, A_t)\Delta t + e^{-\gamma\Delta t}V^*(S_{t+\Delta t})\right]$$

$$= \max_\pi \; \mathbb{E}_{p_\pi}\left[\rho(s_t, A_t)\Delta t + e^{-\gamma\Delta t}\left\{V^*(s_t) + \left(\sum_{i=1}^n \mu^i(s_t, A_t)\frac{\partial V^*(s)}{\partial s_i}\bigg|_{s_t}\right.\right.\right.$$

$$+ \frac{1}{2}\sum_{i=1}^n\sum_{j=1}^n [\sigma(s_t, A_t)\sigma^\top(s_t, A_t)]^{ij}\frac{\partial^2 V^*(s)}{\partial s_i \partial s_j}\bigg|_{s_t}\bigg)\Delta t$$

$$+ \sum_{i=1}^n\sum_{j=1}^m \sigma_j^i(s_t, A_t)\frac{\partial V^*(s)}{\partial s_i}\bigg|_{s_t}(B_{t+\Delta t}^j - B_t^j) + O((\Delta t)^{3/2})\bigg\}\bigg]$$

$$= \max_\pi \; \mathbb{E}_{p_\pi}\left[\rho(s_t, A_t)\Delta t + e^{-\gamma\Delta t}\left\{V^*(s_t) + \left(\sum_{i=1}^n \mu^i(s_t, A_t)\frac{\partial V^*(s)}{\partial s_i}\bigg|_{s_t}\right.\right.\right.$$

$$+ \frac{1}{2}\sum_{i=1}^n\sum_{j=1}^n [\sigma(s_t, A_t)\sigma^\top(s_t, A_t)]^{ij}\frac{\partial^2 V^*(s)}{\partial s_i \partial s_j}\bigg|_{s_t}\bigg)\Delta t + O((\Delta t)^{3/2})\bigg\}\bigg].$$

Simplifying the equation and taking the limit as $\Delta t \to 0$, we have the condition for the optimal value function:

$$V^*(s_t) = \frac{1}{\gamma}\max_\pi \; \mathbb{E}_{p_\pi}\left[\rho(s_t, A_t) + \sum_{i=1}^n \mu^i(s_t, A_t)\frac{\partial V^*(s)}{\partial s_i}\bigg|_{s_t}\right.$$

$$+ \frac{1}{2}\sum_{i=1}^n\sum_{j=1}^n [\sigma(s_t, A_t)\sigma^\top(s_t, A_t)]^{ij}\frac{\partial^2 V^*(s)}{\partial s_i \partial s_j}\bigg|_{s_t}\bigg].$$

### A.3    Convergence of dTD

The existence and uniqueness of the weak solution $V(\cdot)$ to the linear parabolic problem $\partial_t V = TV - V$ with Neumann boundary conditions follow from standard theory for uniformly parabolic equations.

### A.3.1    Proof of Lemma 1

*Proof.* We consider the linear elliptic boundary value problem

$$(\gamma I - \mathcal{L}^\pi)V = \bar{\rho} \quad \text{in } S, \qquad n \cdot D\nabla V = 0 \quad \text{on } \partial S,$$

where

$$(\mathcal{L}^\pi V)(s) = \frac{1}{2}\nabla \cdot (D(s)\nabla V(s)) + b(s) \cdot \nabla V(s), \qquad b := \bar{\mu} - \tfrac{1}{2}\text{div}\, D.$$

**Weak formulation.**    Let $v \in H^1(S)$ be a test function. Multiplying the PDE by $v$ and integrating over $S$ yields

$$\int_S \gamma V v - \int_S (\mathcal{L}^\pi V)\, v = \int_S \bar{\rho}\, v.$$

Using integration by parts for the divergence term and the Neumann condition $n \cdot D\nabla V = 0$ on $\partial S$, we obtain

$$\int_S \gamma V v + \frac{1}{2}\int_S (D\nabla V) \cdot \nabla v - \int_S (b \cdot \nabla V)\, v = \int_S \bar{\rho}\, v.$$

Define the bilinear form $B : H^1(S) \times H^1(S) \to \mathbb{R}$ and the linear functional $f : H^1(S) \to \mathbb{R}$ by

$$B(u, v) := \int_S \gamma u v + \frac{1}{2}\int_S (D\nabla u) \cdot \nabla v - \int_S (b \cdot \nabla u)\, v, \qquad f(v) := \int_S \bar{\rho}\, v.$$

A weak solution is a function $V \in H^1(S)$ such that $B(V, v) = f(v)$ for all $v \in H^1(S)$.

**Boundedness.** Since $D \in L^\infty(S)$ and $b \in L^\infty(S)$, there exists $C > 0$ such that for all $u, v \in H^1(S)$,

$$|B(u,v)| \leq \gamma \|u\|_{L^2} \|v\|_{L^2} + \frac{1}{2} \|D\|_\infty \|\nabla u\|_{L^2} \|\nabla v\|_{L^2} + \|b\|_\infty \|\nabla u\|_{L^2} \|v\|_{L^2}$$

$$\leq C \|u\|_{H^1(S)} \|v\|_{H^1(S)}.$$

Moreover, since $\bar\rho \in L^2(S)$, we have $|f(v)| \leq \|\bar\rho\|_{L^2} \|v\|_{L^2} \leq \|\bar\rho\|_{L^2} \|v\|_{H^1}$, hence $f$ is bounded on $H^1(S)$.

**Coercivity.** Let $v \in H^1(S)$. Using uniform ellipticity of $D$ (i.e., $\xi^\top D \xi \geq \alpha \|\xi\|^2$), we have

$$\frac{1}{2} \int_S (D\nabla v) \cdot \nabla v \geq \frac{\alpha}{2} \|\nabla v\|_{L^2}^2.$$

For the drift term, we use Cauchy–Schwarz and Young's inequality: for any $\varepsilon > 0$,

$$\left| \int_S (b \cdot \nabla v) v \right| \leq \|b\|_\infty \|\nabla v\|_{L^2} \|v\|_{L^2} \leq \varepsilon \|\nabla v\|_{L^2}^2 + \frac{\|b\|_\infty^2}{4\varepsilon} \|v\|_{L^2}^2.$$

Choosing $\varepsilon = \alpha/4$ gives

$$\left| \int_S (b \cdot \nabla v) v \right| \leq \frac{\alpha}{4} \|\nabla v\|_{L^2}^2 + \frac{\|b\|_\infty^2}{\alpha} \|v\|_{L^2}^2.$$

Therefore,

$$B(v,v) = \gamma \|v\|_{L^2}^2 + \frac{1}{2} \int_S (D\nabla v) \cdot \nabla v - \int_S (b \cdot \nabla v) v$$

$$\geq \left( \gamma - \frac{\|b\|_\infty^2}{\alpha} \right) \|v\|_{L^2}^2 + \frac{\alpha}{4} \|\nabla v\|_{L^2}^2.$$

In particular, if $\gamma > \|b\|_\infty^2 / \alpha$, then there exists $c_0 > 0$ such that $B(v,v) \geq c_0 \|v\|_{H^1(S)}^2$ for all $v \in H^1(S)$, i.e., $B$ is coercive on $H^1(S)$.

**Conclusion.** Since $H^1(S)$ is a Hilbert space, $B$ is bounded and coercive, and $f$ is bounded, the Lax–Milgram theorem implies that there exists a unique weak solution $V^\pi \in H^1(S)$ such that $B(V^\pi, v) = f(v)$ for all $v \in H^1(S)$. $\qquad\square$

### A.3.2 Proof of Proposition 2

*Proof.* Let $V^\pi \in H^1(S)$ be the unique weak solution from Lemma 1, and define $e(t) := V(t) - V^\pi$. Since $V^\pi$ satisfies $(\gamma I - \mathcal{L}^\pi)V^\pi = \bar\rho$ and $\partial_t V = -\gamma^{-1}(\gamma I - \mathcal{L}^\pi)V + \gamma^{-1}\bar\rho$, we obtain the error evolution

$$\partial_t e(t) = -\frac{1}{\gamma}(\gamma I - \mathcal{L}^\pi) e(t).$$

We use the weak formulation associated with the bilinear form $B(\cdot, \cdot)$ defined in the proof of Lemma 1. Namely, for any test function $v \in H^1(S)$,

$$\langle \partial_t e(t), v \rangle_{L^2(S)} = -\frac{1}{\gamma} B(e(t), v).$$

Choosing $v = e(t)$ yields the energy identity

$$\frac{1}{2} \frac{d}{dt} \|e(t)\|_{L^2(S)}^2 = -\frac{1}{\gamma} B(e(t), e(t)).$$

By the coercivity estimate established in the proof of Lemma 1, we have for all $w \in H^1(S)$

$$B(w,w) \geq \left( \gamma - \frac{\|b\|_\infty^2}{\alpha} \right) \|w\|_{L^2(S)}^2, \qquad b := \bar\mu - \tfrac{1}{2} \text{div}\, D.$$

Applying this with $w = e(t)$ gives

$$\frac{d}{dt}\|e(t)\|_{L^2(S)}^2 \le -\frac{2}{\gamma}\left(\gamma - \frac{\|b\|_\infty^2}{\alpha}\right)\|e(t)\|_{L^2(S)}^2.$$

Since $\gamma > \|b\|_\infty^2/\alpha$, Grönwall's inequality implies

$$\|e(t)\|_{L^2(S)}^2 \le \exp\left(-\frac{2}{\gamma}\left(\gamma - \frac{\|b\|_\infty^2}{\alpha}\right)t\right)\|e(0)\|_{L^2(S)}^2,$$

and hence

$$\|V(t) - V^\pi\|_{L^2(S)} \le \exp(-\lambda t)\,\|V(0) - V^\pi\|_{L^2(S)}, \qquad \lambda := \frac{1}{\gamma}\left(\gamma - \frac{\|b\|_\infty^2}{\alpha}\right) > 0.$$

$\square$

# B Implementation Details

## B.1 Algorithm

The procedures for policy evaluation are summarized in Algorithm 1.

---
**Algorithm 1** Policy evaluation with dTD

---
**Input:** policy $\pi$
**Output:** $V_\theta$
  Initialize value function $V_\theta$ with random parameter $\theta$
  **for** each training step **do**
    Initialize buffer $\mathcal{D} = \emptyset$ and initial state $s_0$
    **for** each environment step **do**
      $a_t \sim \pi(\cdot|s_t)$
      $s_{t+\Delta t} \sim p(\cdot|s_t, a_t)$
      $\mathcal{D} \leftarrow \mathcal{D} \cup (s_t, a_t, s_{t+\Delta t}, \rho_t)$
    **end for**
    **for** each update step **do**
      Sample a batch of $D$ random transitions from $\mathcal{D}$
      $\bar{\theta} \leftarrow \theta$
      $y_d \leftarrow -\rho_t^d - \gamma V_{\bar{\theta}}(s_{t+\Delta t}^d)$
      $\text{pred}_d \leftarrow \sum_{i=1}^{n} \frac{(s_{t+\Delta t}^{d,i} - s_t^{d,i})}{\Delta t} \left.\frac{\partial V_\theta(s)}{\partial s_i}\right|_{s_t^d} + \frac{1}{2}\sum_{i=1}^{n}\sum_{j=1}^{n} \frac{(s_{t+\Delta t}^{d,i} - s_t^{d,i})(s_{t+\Delta t}^{d,j} - s_t^{d,j})}{\Delta t}\left.\frac{\partial^2 V_\theta(s)}{\partial s_i \partial s_j}\right|_{s_t^d}$
      Update parameter $\theta$ using gradient descent method
      $\theta \leftarrow \text{argmin}_\theta \frac{1}{D}\sum_{d=1}^{D}(y_d - \text{pred}_d)^2$
    **end for**
  **end for**

---

## B.2 Efficient Computation of the dTD Loss

Since equation (10) involves the Hessian, it may seem that $O(n^2)$ (where $n$ is the dimension of the observation space) computations are required. However, by rearranging the order of calculations, such as using

$$\left\langle \Delta s_t, \left.\frac{\partial^2 V(s)}{\partial s^2}\right|_{s_t} \Delta s_t \right\rangle = \left\langle \Delta s_t, \left.\frac{\partial}{\partial s}\left\langle \frac{\partial V}{\partial s}, \Delta s_t \right\rangle\right|_{s_t} \right\rangle,$$

we can avoid directly calculating the Hessian and achieve a computation complexity of $O(n)$.

## B.3 Hyperparameters for PPO

The search space of the hyperparameters is summarized in Table 2. The values chosen finally are summarized in Tables 3.

Table 2: Hyperparameter search space

| Hyperparameter | Search Space |
| --- | --- |
| environment steps per update (number of parallel environment: 64) | $\{8, 16, 32\}$ |
| number of epochs per update | range(5, 20) |
| minibatch size | $\{256, 512\}$ |
| learning rate | $\log(\text{interval}(1e-6, 5e-3))$ |
| normalize advantage | $\{\text{True}, \text{False}\}$ |
| gae lambda | interval(0.8, 0.9999) |
| clip range | interval(0.0, 0.9) |
| entropy coefficient | interval(0.0, 0.3) |
| value loss weight | interval(0.0, 1.0) |
| mixture raio $\beta$ | interval(0.0, 1.0) |

Table 3: Best hyperparameters for PPO with TD and $\beta$-dTD across environments

| Hyperparameter | TD | | | | $\beta$-dTD | | | |
| --- | --- | --- | --- | --- | --- | --- | --- | --- |
| | Hopper | Halfcheetah | Ant | Humanoid | Hopper | Halfcheetah | Ant | Humanoid |
| environment steps/update | 32 | 16 | 8 | 16 | 32 | 8 | 32 | 16 |
| epochs/update | 7 | 5 | 11 | 15 | 19 | 9 | 16 | 10 |
| minibatch size | 512 | 256 | 512 | 512 | 256 | 256 | 256 | 256 |
| learning rate | 1.18e-3 | 5.93e-4 | 3.71e-4 | 1.40e-3 | 3.52e-4 | 3.29e-4 | 7.94e-5 | 1.08e-3 |
| normalize advantage | False | False | False | False | False | False | False | False |
| GAE lambda | 0.886 | 0.833 | 0.935 | 0.999 | 0.998 | 0.908 | 0.805 | 0.888 |
| clip range | 0.439 | 0.268 | 0.425 | 0.063 | 0.075 | 0.040 | 0.520 | 0.713 |
| entropy coefficient | 0.121 | 0.018 | 0.162 | 0.021 | 0.046 | 0.011 | 0.133 | 0.002 |
| value loss weight | 0.049 | 0.513 | 0.711 | 0.091 | 0.675 | 0.268 | 0.274 | 0.054 |
| mixture ratio $\beta$ | — | — | — | — | 0.572 | 0.742 | 0.241 | 0.332 |

## B.4 Details of the A2C Implementation

### B.4.1 Learning Curves

### B.4.2 Hyperparameters for A2C

The search space of the hyperparameters is summarized in Table 4. The values chosen finally are summarized in Tables 5.

Table 4: Hyperparameter search space

| Hyperparameter | Search Space |
| --- | --- |
| environment steps per update (number of parallel environment: 64) | $\{8, 16, 32\}$ |
| number of epochs per update | range(5, 20) |
| minibatch size | $\{256, 512\}$ |
| learning rate | $\log(\text{interval}(1e-6, 5e-3))$ |
| normalize advantage | $\{\text{True}, \text{False}\}$ |
| gae lambda | interval(0.8, 0.9999) |
| entropy coefficient | interval(0.0, 0.3) |
| value loss weight | interval(0.0, 1.0) |
| mixture raio $\beta$ | interval(0.0, 1.0) |

## B.5 Computing Infrastructure and Reproducibility

**Computing infrastructure** Experiments were conducted on a machine with four NVIDIA Tesla V100 GPUs (32GB each) and an Intel Xeon E5-2698 v4 CPU. Although all experiments can be executed on a single GPU, multiple GPUs were used to run independent trials in parallel for efficiency.

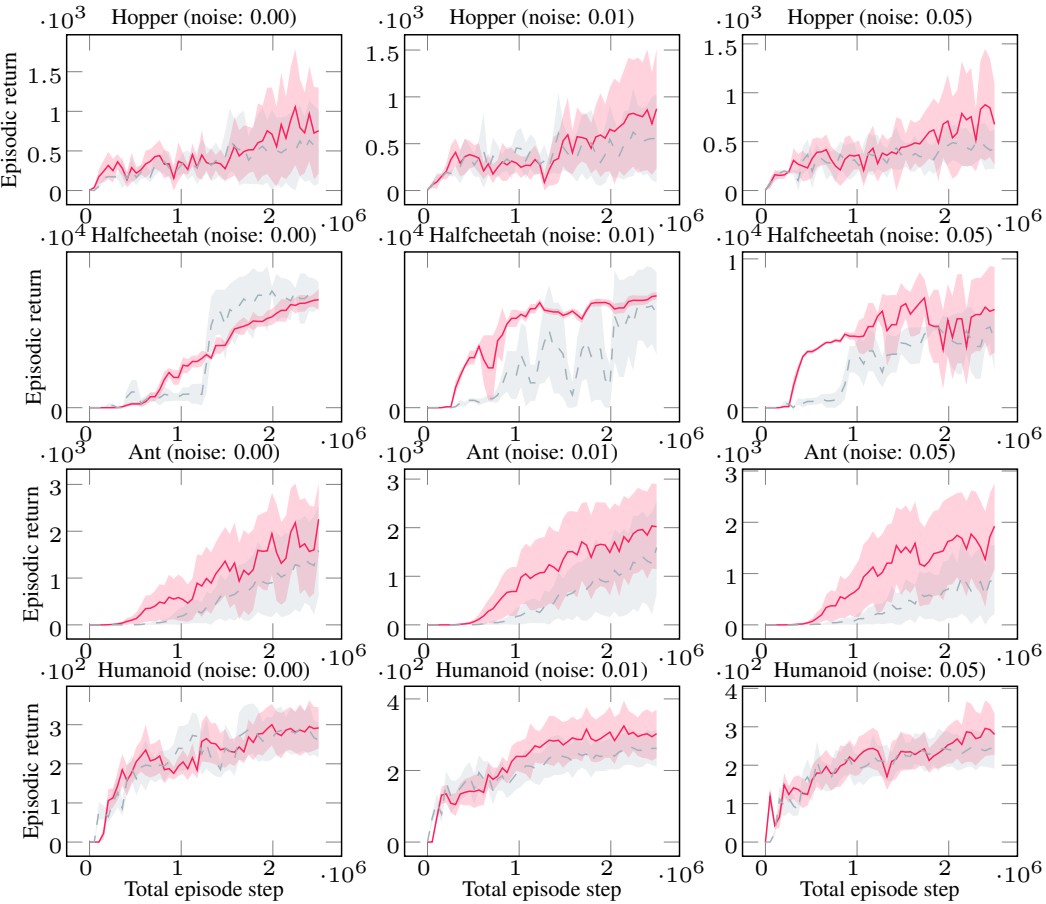

Figure 3: Performance of TD, and $\beta$-dTD on continuous control benchmark. These results were obtained using the A2C algorithm. Each column corresponds to different noise levels (coef = 0.00, 0.01, 0.05), and each row corresponds to different environments. The tuned $\beta$ values for $\beta$-dTD were 0.24, 0.60, 075, 0.47 in Hopper, HalfCheetah, Ant, and Humanoid, respectively.

Table 5: Best hyperparameters for A2C with TD and $\beta$-dTD across environments

| Hyperparameter | TD | | | | $\beta$-dTD | | | |
|---|---|---|---|---|---|---|---|---|
| | Hopper | Halfcheetah | Ant | Humanoid | Hopper | Halfcheetah | Ant | Humanoid |
| environment steps/update | 32 | 16 | 8 | 16 | 16 | 8 | 8 | 16 |
| epochs/update | 7 | 5 | 11 | 15 | 16 | 9 | 12 | 19 |
| minibatch size | 512 | 256 | 512 | 512 | 512 | 256 | 256 | 512 |
| learning rate | 1.18e-3 | 5.93e-4 | 3.71e-4 | 1.40e-3 | 1.88e-6 | 3.52e-4 | 2.90e-6 | 6.83e-7 |
| normalize advantage | False | False | False | False | False | False | False | True |
| GAE lambda | 0.886 | 0.833 | 0.935 | 0.999 | 0.890 | 0.950 | 0.960 | 0.827 |
| entropy coefficient | 0.121 | 0.018 | 0.162 | 0.021 | 0.031 | 5.38e-6 | 0.094 | 0.046 |
| value loss weight | 0.049 | 0.513 | 0.711 | 0.091 | 0.825 | 0.602 | 0.457 | 0.546 |
| mixture ratio $\beta$ | — | — | — | — | 0.244 | 0.598 | 0.750 | 0.467 |

**Training time** Hyperparameter tuning typically took 6–9 hours depending on the environment. Training time for the final runs depended on the environment and ranged from 10 to 60 minutes.

**Reproducibility** All experiments were conducted with the random seed fixed in the training scripts. However, MuJoCo (accessed via Brax) uses its own internal random seed that is not directly controllable, so full determinism cannot be ensured. The code is available at `https://github.com/4thhia/differential_TD` for reproducibility.

