# OpenReview forum: "A Temporal Difference Method for Stochastic Continuous Dynamics"
_NeurIPS.cc/2025/Conference — NeurIPS 2025 poster_

### Official Review · Reviewer_TdDL · 2025-06-09

**Clarity:** 4
**Significance:** 3
**Originality:** 3
**Rating:** 4
**Confidence:** 4

**Summary:**

This paper introduces a Reinforcement Learning algorithm that updates the value network using the stochastic HJB equation, thus bridging discrete-time and continuous-time insights. This method is tested on the MuJoCo environments and outperforms PPO.

Unfortunately, although the experimental results are promising, I am concerned about the author's familiarity with HJB equations and important characteristics of the problem. I am willing to increase my score if the authors focus on making their work more grounded in the literature on viscosity solutions and continuous-time policy iteration in their rebuttal.

**Questions:**

* In the abstract, "Bridging stochastic optimal control and model-free reinforcement learning", what is the difference?
* In Appendix A.3, where you use Ito's lemma, you are missing $\Delta t$ in some parts of the equation.
* How is Proposition 2 used in the numerical experiments? How are $L_v$ and $L_\sigma$ computed?

**Ethical Concerns:**

["NO or VERY MINOR ethics concerns only"]

**Final Justification:**

Through the rebuttal, the authors have addressed my concern with the theoretical grounding of their paper, namely that the value function is not Lipschitz and smooth. This assumption has been relaxed, and the condition for contraction of the HJB operator has been replaced with an inequality that is grounded in more realistic assumptions, thus significantly improving the mathematical foundation of the paper.

**Limitations:**

* I appreciate that the authors acknowledge some limitations (e.g., instability, off-policy inapplicability, large $\gamma$) of their work.
* As mentioned above, I suggest that the authors conduct an exploration of regions where $V$ isn't first or second-order differentiable.

**Quality:**

3

**Strengths And Weaknesses:**

**Strengths**

* The paper is well-written and mathematically easy to follow. I followed the proof of Proposition 1 and the details on the HJB equation and believe both are correct. Appendix A.1 is correct and carefully grounded in the literature on stochastic differential equations.
* Although Proposition 2 is based on problematic assumptions, I went through the proof and believe it is correct.
* The derivation from the Bellman equation to the HJB to the dTD update is clear and well-motivated.
* The experimental results seem promising. PPO with $\beta$-TD significantly outperforms PPO in six out of twelve environments and shows comparable performance in the remaining environments (except half-cheetah)

**Weaknesses**

* I find Proposition 2 problematic. How are the assumptions verified? Assumption (i) is strong and may not hold in general. This undermines the generality of the contraction proof. $V$ will be, at best, locally Lipschitz in some sub-domains. Furthermore, it is not scalable: as n and m increase, $\gamma$ needs to increase as well, which can severely limit the model's ability to consider long-horizon planning. How are $L_v$ and $L_\sigma$ computed in numerical experiments?
* In general, the value function will not have continuous first or second order derivatives - rather, it will be a viscosity solution (indeed, the study of viscosity solutions was motivated by the fact that the value function is not generally differentiable). For example, in LQR settings, the value function is not differentiable at the origin. I highly recommend that the authors read the paper [1] on neural network-based methods of mitigating these issues.
* Fundamentally, the proposed method combines discrete-time and continuous-time solutions, which I believe is problematic. The proposed algorithm is based on PPO and thus uses the policy gradient (PG) theorem [2], which is derived in the discrete-time domain. If the authors want to use a policy gradient algorithm, they need to build their method based on the continuous-time variant of PG, for instance, pathwise policy gradient estimators [3].
* Important baselines are missing: (1) Soft Actor-Critic [4], (2) HJ DQN [5].

[1] Yiming Meng et al. "Physics-Informed Neural Network Policy Iteration: Algorithms, Convergence, and Verification." ICML 2024.

[2] Richard Sutton et al. "Policy gradient methods for reinforcement learning with function approximation." NeurIPS 1999.

[3] Remi Munos. "Policy gradient in continuous time." JMLR (2006).

[4] Tuomas Haarnoja. "Soft Actor-Critic: Off-Policy Maximum Entropy Deep Reinforcement Learning with a Stochastic Actor." ICML 2018.

[5] Jeongho Kim et al. "Hamilton–Jacobi Deep Q-Learning for Deterministic Continuous-Time Systems with Lipschitz Continuous Controls." JMLR (2021)

---

> ### Author Rebuttal · Authors · 2025-07-31
>
> We thank the reviewer for their detailed feedback and for engaging with our work.
> Despite the favorable evaluation regarding the quality (good), clarity (excellent), significance (good), and originality (good) of the paper, seemingly some miscommunication and misunderstanding led to the current overall rating.
> We believe the answers below would be a thorough clarification.
>
> # Response to Limitations Raised by the Reviewer
> ## How to compute Lipschitz coefficient of V?
> As shown in Appendix B.3, we do not need to compute the Lipschitz coefficient of $V$. This requirement appears only in the theoretical analysis to ensure convergence. We believe this limitation is acceptable, as many widely adopted methods, such as SGD or Nesterov's accelerated methods (including variants like Adagrad), make similar assumptions solely for theoretical purposes.
> There, as is the case with our work, we do not need to know the Lipschitz constant in practice.
> For example, typically one does not determine the step size of SGD by explicitly computing the Lipschitz constant of the objective function.
>
> ## Differentiability of $V$
> As the reviewer pointed out, the viscosity solution of the HJB equation is continuous but not necessarily everywhere differentiable. However, the differentiability requirement in our analysis applies not to the viscosity solution itself, but to the function approximator $V_\theta$. Neural networks can be designed to be twice differentiable and can approximate any continuous function including viscosity solutions.
>
> ## Continuous time policy gradient
> We do not combine discrete-time and continuous-time formulations in the inconsistent way that the reviewer seems to have in mind.
> We guess some misinterpretation of the policy gradient theorem resulted in the reviewer's concern.
> The policy gradient theorem was originally introduced in discrete-time RL, but the derivation is not specific to discrete time. The core idea relies on the log-derivative trick:
> $$
> \begin{aligned}\nabla_\theta \mathbb E_{x\sim P_{\theta}(x)}[f(x)] &= \nabla_\theta\int f(x)P_\theta(dx) \\ &=\int f(x)\nabla_\theta P_\theta(dx) \\ &=\int f(x)\frac{\nabla_\theta P_\theta(x)}{P_\theta(x)}P_\theta(dx) \\ &= \mathbb E_{x\sim P_\theta(x)}\left[f(x)\frac{\nabla_\theta P_\theta(x)}{P_\theta(x)}\right] \\ &= \mathbb E_{x\sim P_\theta(x)}\left[f(x)\nabla_\theta\log{P_\theta(x)}\right] \\ \end{aligned}
> $$
> This formulation applies equally in both discrete and continuous-time settings. In discrete RL, we take $f \rightarrow \sum_{t=0}^{\infty} \gamma^t r(s_t, a_t)$; in continuous RL, we use $f \rightarrow \int_{0}^{\infty} e^{-\gamma t} \rho(s_t, a_t) dt$. When applying the method to MuJoCo environments, which operate in discrete simulation steps, we discretize the continuous-time formulation accordingly. This discretization ultimately yields the same form as standard discrete-time policy gradient methods.
>
> The cited paper by Munos [3] proposes a method for improving policy gradients in continuous time, but it is not a formulation of the baseline continuous time policy gradient itself.
>
> ## Missing baselines
> Soft Actor-Critic is an off-policy method, and as noted several times in the paper, our proposed approach is designed for on-policy algorithms such as PPO and A2C.
>
> Direct comparison to HJ DQN [5] is very meaningful. It focuses on improving the computation of the $\arg\max_a Q(s, a)$ operation, while our method directly modifies the TD update itself. These approaches target fundamentally different components of the RL algorithm, and comparing them is like comparing apples and oranges. In fact, our method could in principle be combined with HJ DQN—for example, by incorporating our proposed TD update into Equation (22) of [5]. The two are not competing methods but rather orthogonal in design.
>
> # Answer to questions
> ## Difference between stochastic control and RL
> In stochastic control, the HJB equation is typically solved analytically under various assumptions. In contrast, RL solves the Bellman equation in a data-driven manner without such assumptions. In continuous RL, there has been interest in solving the HJB equation in a data-driven manner similar to RL. However, existing approaches typically require either access to the true dynamics or an explicit model estimation step, creating a gap from how RL is practiced. In this work, we demonstrate that it is indeed possible to solve stochastic control problems in a fully model-free, RL-style manner.
>
> We have also included experimental results for A2C and the model-known setting as part of our additional analysis. For further details, we kindly refer you to our responses to the other reviewers.

---

> ### Author Response · Authors · 2025-08-04
> **Clarification of Assumptions and Scope of Our Study**
>
> Thank you very much for the effort for the discussion.
> We appreciate the comments for further streamlining the presentation of the work.
> Meanwhile, we feel some miscommunication remains regarding our assumptions and goal.
> # Lipschitz constant and differentiability of $V_\theta$
> There seems to be a miscommunication regarding the assumptions made on the true value function $V$ and the function approximator $V_\theta$. We clarify these separately. Importantly, we believe none of the assumptions we make here are strong.
> ## Assumptions
> - We only assume that $V$ is a viscosity solution, which is always continuous. We do not assume $V$ to be differentiable or Lipschitz continuous.
> - For $V_\theta$, a neural network, we assume differentiability and Lipschitz continuity. Neural networks are differentiable by design. As for Lipschitz continuity, neural networks are typically Lipschitz continuous. As is well known, if $f = f_2 \circ f_1$, then $L_f \leq L_{f_2} \cdot L_{f_1}$.
> Since neural networks are composed of affine maps (e.g., $Wx + b$) and Lipschitz activation functions (e.g., ReLU, Sigmoid), the entire network remains Lipschitz continuous.
>
> ## Gap between $V$ and $V_\theta$
>
> We find no issue in approximating a viscosity solution, which may be non-differentiable but is always continuous, using a differentiable neural network.
> The value function, i.e., the solution to the Bellman equation, is often not even continuous, yet standard deep RL methods approximate it using neural networks.
> This is a common gap usually regarded as harmless; questioning it involves the foundation of deep RL, which is clearly out of scope of the paper.
>
> ## Limitation
> A conceivable limitation of the method is that, to ensure fully stable learning in practice, one might need to explicitly compute the Lipschitz constant $L_{V_\theta}$.
> However, we believe this is an acceptable limitation.
> In fact, widely-accepted methods such as SGD and Adagrad also rely on the Lipschitz constant in theoretical analyses, e.g., for determining step sizes.
> However, practitioners almost never compute the Lipschitz constant for such a purpose.
> If one really wishes to compute it for some reason, there is a method to compute its upper bound [R1].
>
> [R1] Scaman, K., \& Virmaux, A. (2018). Lipschitz regularity of deep neural networks: analysis and efficient estimation. In Advances in Neural Information Processing Systems (NeurIPS 2018).
>
> # Baselines
> We appreciate the suggestion.
> While comparison to off-policy methods might be an interest from a practically-oriented point of view, we do not think that it strengthens the main claim of the paper.
> We acknowledge that in many RL studies, the methods are compared against SAC; this is because they propose standalone RL algorithms intended to be competitive with SAC.
> In contrast, our aim is to demonstrate that HJB-based TD methods, which were previously considered incompatible with deep RL, can in fact be implemented within standard deep RL pipelines.
> To this end we improve existing on-policy algorithms in a general and principled manner.
>
> Our proposal, dTD, is a general technique designed to improve on-policy methods.
> Therefore the important comparison is between the same base on-policy methods with and without dTD.
> It may or may not outperform SAC; it mostly depends on the base method and the contribution of the use of dTD cannot be assessed from such a comparison.
>
> # $\gamma$ in Proposition 2
> We acknowledge that referring to a "large $\gamma_{\text{continuous}}$" as the subject of the assumption was misleading and, in fact, unnecessary. As shown in Proposition 2, contraction can be guaranteed either by a large $\gamma_{\text{continuous}}$ or by small Lipschitz constants of dynamics, $L_\mu$ and $L_\sigma$.
> We should have focused on the latter for better understanding the assumption.
> The actual limitation is not the need for a large $\gamma_{\text{continuous}}$ but rather lies in the fact that *the method should be applied to systems with sufficiently smooth dynamics* (i.e., small Lipschitz constants of $\mu$ and $\sigma$).
> The Lipschitz continuity of $\mu$ and $\sigma$ is naturally required from the existence of solutions to the SDEs, so it is not an additional assumption.
> Moreover, when the dynamics are smooth enough, faster convergence than standard TD methods can be expected.
>
> In practice, our method works well with proper discount factors $\gamma_{\text{discrete}}$ in the range [0.8, 0.99], as demonstrated in our experiments (see Appendix B.2).
>
> Here, just to clarify, "large $\gamma$" should refer to the continuous-time discount rate $\gamma_{\text{continuous}}$, not to the discrete one.
> Since $\gamma_{\text{discrete}} = e^{-\gamma_{\text{continuous}} \Delta t}$, even a large $\gamma_{\text{continuous}}$ yields a valid $\gamma_{\text{discrete}} \in (0, 1)$, which causes no problems conceptually.
>
> We will revise the description of Proposition 2 to make the dependency on $L_\mu$ and $L_\sigma$ explicit.

---

> > ### Comment · Reviewer_TdDL · 2025-08-05
> >
> > Thank you for your additional clarifications. I appreciate the authors’ willingness to engage deeply with the concerns. The paper presents interesting ideas, and I believe the theoretical development has potential. However, the theoretical aspects still seem misaligned with the implementation.
> >
> > Proposition 2 presents a contraction result that relies crucially on condition (iv), involving the Lipschitz constant $L_V$ of the value function. While it is true that neural networks are Lipschitz continuous by construction, this constant can be large and does not necessarily reflect the true value function’s regularity, particularly if $V$ is not Lipschitz (which is entirely possible in the viscosity solution setting). While it is true that neural networks are used to approximate functions that lack smoothness, the situation here is distinct because *the theoretical results rely on the Lipschitz continuity* (and $\beta_V$-smoothness) of $V_\theta$ to ensure contraction. The rebuttal frames this as a general issue in deep RL, but in that broader context, contraction results are often not claimed at all.
> >
> > **My condition on increasing the score** is if the authors can replace condition (iv) in Proposition 2 with a different inequality that does not depend on $L_V$. For instance, the authors can consider the supremum of $V$ along the state space $S$ instead, which is assumed to exist in many theoretical works. Can the authors also provide a proof? If so, then I trust that the authors will remove all the occurrences of $L_V$ in the final version of the paper.

---

> ### Author Response · Authors · 2025-08-07
> **Proof of contraction without Lipschitz constant of V_θ**
>
> Thank you for this suggestion and for the deep engagement in the discussion. We can prove the contraction regardless of the Lipschitz continuity of $V_\theta$. In the revised manuscript, we replace the original assumptions and proof with those presented here, which we find preferable.
>
> The idea of the proof is to shift $\nabla$ applied to $V$ to the other terms by integration by parts, and then eliminate the extra terms using Gauss's divergence theorem together with the $H^{-1}$ norm, which is the norm on the dual space of the Sobolev space. For $\nabla^2$, this operation is simply repeated twice.
>
> The new set of assumptions are:
> - $S$ is bounded (as specified in the paper);
> - $\mu,\sigma\in C_b^2(S)$ (twice continuously differentiable, with each function and its first two derivatives bounded); and
> - $V_\theta\in L^2(S)$ (holds automatically, since $V_\theta$ is continuous by design and $S$ is bounded).
>
> Note that assuming $\mu,\sigma\in C_b^2(S)$ is standard in SDE theory.
>
> The $H^{-1}(S)$ norm is defined by
>
> $$\lVert f\rVert_{H^{-1}(S)}:=\sup_{\phi\in H_0^1(S),\lVert\phi\rVert_{H_0^1(S)}=1}\int_S f(x) \phi(x) dx,$$
>
> where $H_0^1(S)$ is the Sobolev space of functions vanishing on ∂$S$ with square integrable first derivatives (see [R2] for details).
>
> We focus on the 1D case for space reasons; each technique extends naturally to higher dimensions. We write as $\nabla := \dfrac{d}{ds},\ \nabla^2 := \dfrac{d^2}{ds^2}$ and $w:=V_2^\pi-V_1^\pi$ for notational brevity.
>
> First, the original proof carries over unchanged—only now we measure everything in the $H^{-1}(S)$ norm:
> $$\lVert TV_2^\pi - TV_1^\pi\rVert_{H^{-1}}=\lVert Tw\rVert_{H^{-1}}=\left\lVert \dfrac{1}{\gamma} \mathbb{E}\left[\mu\nabla w+\frac{1}{2}\sigma^2\nabla^2 w\right]\right\rVert_{H^{-1}(S)}$$
> $$\leq \dfrac{1}{\gamma} \mathbb{E}\left[\left\lVert \mu\nabla w+\frac{1}{2}\sigma^2\nabla^2 w\right\rVert_{H^{-1}(S)}\right] \leq \dfrac{1}{\gamma} \mathbb{E}\left[\left\lVert \mu\nabla w\right\rVert_{H^{-1}(S)}+\frac{1}{2}\left\lVert \sigma^2\nabla^2 w\right\rVert_{H^{-1}(S)}\right]$$
> The first inequality holds, by convexity of the norm and Jensen’s inequality. We then transform the first term via integration by parts.
> $$\left\lVert \mu\nabla w\right\rVert_{H^{-1}(S)}=\sup_{\phi\in H_0^1(S),\lVert\phi\rVert_{H_0^1(S)}=1}\int_S \mu (\nabla w) \phi\ dx$$
> $$=\sup_{\phi\in H_0^1(S),\lVert\phi\rVert_{H_0^1(S)}=1} \left(\int_S \nabla (\mu w \phi)\ dx-\int_S w \nabla (\mu \phi)\ dx\right)\ \ \cdots(1)$$
> $$=\sup_{\phi\in H_0^1(S),\lVert\phi\rVert_{H_0^1(S)}=1} \int_S w \nabla (\mu \phi)\ dx\ \ \cdots(2)$$
> $$=\sup_{\phi\in H_0^1(S),\lVert\phi\rVert_{H_0^1(S)}=1} \lVert \nabla (\mu \phi)\rVert_{H_0^{1}(S)} \int_S w \frac{\nabla (\mu \phi)}{\lVert \nabla (\mu \phi)\rVert_{H_0^{1}(S)}}\ dx\ \ \cdots(3)$$
> $$\leq \left(\sup_{\phi\in H_0^1(S),\lVert\phi\rVert_{H_0^1(S)}=1} \lVert \nabla (\mu \phi)\rVert_{H_0^{1}(S)}\right) \lVert w\rVert_{H^{-1}(S)}\ \ \cdots(4)$$
> (1) is integration by parts. In (1) $\to$ (2),Gauss's divergence theorem turns the first term into a boundary integral, which vanishes on ∂$S$ (because $\phi\in H_0^1(S)$).
> The second term’s sign flips since we take supremum over $\phi$ on the unit ball. (this is why the $H^{-1}$ norm is defined without taking absolute values).
> For (2) $\to$ (3), we need ∇$(\mu\phi)\in H_0^1(S)$; this is ensured by taking $\phi\in C_c^\infty(S)\subset H_0^1(S)$. Since $C_c^\infty(S)$ is dense in $H_0^1(S)$, restricting the supremum to $C_c^\infty(S)$ does not change its value.
> Inequality (4) follows by taking supremum over $\theta(x):=\frac{\nabla (\mu \phi)}{\lVert \nabla (\mu \phi)\rVert_{H_0^{1}(S)}}$ . (i.e., $\lVert\theta\rVert_{H_0^1(S)}=1$).
>
> As for the coefficient,
>
> $$
> \lVert \nabla (\mu \phi)\rVert_{H_0^{1}(S)} = \lVert \nabla (\mu \phi)\rVert_{L^{2}(S)} + \lVert \nabla^2 (\mu \phi)\rVert_{L^{2}(S)}
> $$
> $$
> \leq \left(\lVert \nabla \mu\rVert_{L^{\infty}(S)} \lVert \phi\rVert_{L^{2}(S)} + \lVert \mu\rVert_{L^{\infty}(S)} \lVert \nabla \phi\rVert_{L^{2}(S)}\right) + \left(\lVert \nabla^2 \mu\rVert_{L^{\infty}(S)} \lVert \phi\rVert_{L^{2}(S)} + \lVert \nabla \mu\rVert_{L^{\infty}(S)} \lVert \nabla \phi\rVert_{L^{2}(S)} + \lVert \mu\rVert_{L^{\infty}(S)} \lVert \nabla^2 \phi\rVert_{L^{2}(S)} \right)
> $$
>
> These terms are automatically bounded as $\mu\in C_b^2, \phi\in C_c^\infty, \lVert\phi\rVert_{H_0^1}=1$, and $S$ is bounded. Thus,
>
> $$\left\lVert \mu\nabla(V_2^\pi - V_1^\pi)\right\rVert_{H^{-1}(S)} \leq C_1 \left\lVert V_2^\pi - V_1^\pi\right\rVert_{H^{-1}(S)}.$$
> By repeating the integration-by-parts argument twice, we obtain
>
> $$\left\lVert \sigma^2 \nabla^2(V_2^\pi - V_1^\pi)\right\rVert_{H^{-1}(S)} \leq C_2 \left\lVert (V_2^\pi - V_1^\pi)\right\rVert_{H^{-1}(S)}.$$
> The constants $C_1$ and $C_2$ become smaller as the dynamics are smoother; that is, as the first and second derivatives of $\mu$ and $\sigma$ become smaller.
>
> [R2] L.C.Evans, Partial Differential Equations, Graduate Studies in Mathematics, 1997.

---

> ### Comment · Reviewer_TdDL · 2025-08-07
>
> Huge thanks to the authors for providing a proof in a very short amount of time! I went through the proof in detail, and overall, it looks convincing.
>
> Just a few minor errors:
> The first line for $\lVert TV_2^\pi - TV_1^\pi\rVert_{H^{-1}}$ should be
> $$\lVert TV_2^\pi - TV_1^\pi\rVert_{H^{-1}}=\lVert Tw\rVert_{H^{-1}}=\left\lVert \dfrac{1}{\gamma} \mathbb{E}\left[\mu\cdot\nabla w+\frac{1}{2}\sigma^2\cdot\nabla^2 w\right]\right\rVert_{H^{-1}(S)}$$
> You missed the dot product (and matrix dot product for the $\sigma^2$ term).
> This means (2) should be
> $$\sup_{\phi\in H_0^1(S),\lVert\phi\rVert_{H_0^1(S)}=1} \int_S w \nabla \cdot (\mu \phi)\ dx,$$
> which still bounds $\mu$ due to the Holder inequality. Your bound now also contains the $L_2$ and $L_\infty$-norms of $\nabla(\nabla\cdot\mu)$, both of which exists since $\mu\in C_b^2(S)$.
>
> That said, I agree that the paper has shown significant improvement during this discussion period, both theoretically and experimentally. I believe a contraction property in HJB/RL is an impactful theoretical contribution, even if it requires $\gamma$ to be large, and the updated theorem grounds it in realistic assumptions. I increase my score from 2 to 4. It was a pleasure reviewing your paper.

---

> > ### Author Response · Authors · 2025-08-07
> >
> > Thank you very much for your constructive and thoughtful comments throughout the discussion. We truly appreciate the time and effort you put into the discussion!

---

### Official Review · Reviewer_jPHL · 2025-06-11

**Clarity:** 2
**Significance:** 4
**Originality:** 3
**Rating:** 5
**Confidence:** 5

**Summary:**

This paper proposes an alternative algorithm to TD learning, called differential TD (dTD), for continuous-time, stochastic RL problems. Contrary to existing approaches from the optimal control literature, this algorithm is model-free (sample-based) like TD learning, but exploits the continuity of the value function. The authors demonstrate the practical potential of this approach, when mixed with classical TD iterations.

**Questions:**

- The assumptions of Proposition 2 are appropriately discussed in the paper. It is indeed known that $V$ has in general little regularity in the deterministic case. However I was wondering if something more could be said in the stochastic setting: the noise in the dynamics is usually seen as having a regularizing effect on the value function. See for example [Fleming and Rishel, *Deterministic and Stochastic Optimal Control*, 2012], where the Brownian motion makes the value function at least C^2 in the space variable.
- In Eq. 8, a term $O(\Delta t^{3/2})$ is missing from the rhs of the equation.
- Is it possible to exhibit a toy example where exploiting the continuity of the value function is key, and makes dTD perform significantly better than plain TD? Did the experiments that you led give you any insight on the type of problems where dTD is expected to perform best?
- I understand that mixing dTD with TD is better in terms of stability, however I believe that dTD (with $\beta=1$) should be reported as well in the plots, even as a baseline.
- Can you provide some more details on the parametric representation of $V$? Is it exactly the same for the TD, dTD and naive dTD algorithms? Is it adapted to the different problem instances?
- Can you please clarify the exact difference between dTD and naive dTD, and why their performance is so different?

**Ethical Concerns:**

["NO or VERY MINOR ethics concerns only"]

**Final Justification:**

The authors have answered most of the my questions. While certain weaknesses may remain in the paper, I believe it is of interest for the community, and could pave the way for future interesting work. Therefore I update my score from 4 to 5 and recommend to accept this paper.

**Limitations:**

Limitations concerning stability issues and the large discount factor regime are appropriately mentioned in the paper.

**Quality:**

3

**Strengths And Weaknesses:**

This paper, well-written and theoretically-grounded, tackles an interesting problem, in the relatively under-studied continuous setting in RL. The proposed algorithm is sample-based, hence compatible with RL problems. While it is limited to fixed-policy TD-based approaches, the paper shows with PPO how it can be applied to optimize the policy as well. The experiments are thoroughly led, with interesting discussions on the influence of the different parameters. The limitations are mainly related to the experimental comparison with the plain TD algorithm, where the actual advantage of dTD is difficult to assess in all situations. Here are more detailed limitations (see also questions below):
- While the contraction analysis provided in Proposition 2 is important, a consistency result as $\Delta t \rightarrow 0$ is missing from the analysis. When dTD is applied with a finite $\Delta t$, can we say anything about the error on the value function or the Bellman error?
- In the experiments, the naive dTD and dTD algorithms have very different behaviors. However, since the  loss function is mentioned to be the MSE, I don’t understand how changing the target and prediction would change anything (up to a scaling factor of $\gamma$)? This is the most confusing detail of the paper, and I think the precise difference between the two updates should be made much clearer.
- TD is an algorithm designed for policy evaluation. Yet in the experiments, only PPO is tested, while TD is only a sub-element of it. Is it possible to assess the performance of dTD on policy evaluation only, where the distance to $V^{\pi}$ can be directly evaluated? Maybe the influence of the different parameters would be easier to judge on this specific sub-problem.
- As discussed in the paper, the optimal $\beta$ depends on the problem. However, it is quite difficult to judge whether finding that some $\beta >0$ performs better than $\beta=0$ on some problems is indeed significant, or is only due to the fact that we are optimizing one more free parameter.

---

> ### Author Rebuttal · Authors · 2025-07-31
>
> Thank you for your thoughtful and insightful questions. Below, we address them in detail:
>
> # Response to Limitations Raised by the Reviewer
> ## On the analysis of finite difference
> As shown in Equation (7), the finite-difference approximation $\mathbb{E}\left[\frac{S_{t+\Delta t} - S_t}{\Delta t}\right]$
> deviates from the true drift term $\mu$ by an error of order $O(\Delta t^{3/2}).$ This deviation directly affects the contraction coefficient in Proposition 2, as it introduces a corresponding perturbation to the Lipschitz constant of the dynamics.
>
> As a side note, in practice, the algorithm uses the empirical finite difference
> $\frac{S_{t+\Delta t} - S_t}{\Delta t}$ instead of its expectation. This can be decomposed as:
> $$
> \frac{S_{t+\Delta t} - s_t}{\Delta t} = \mathbb{E}\left[\frac{S_{t+\Delta t} - s_t}{\Delta t}\right] + \left(\frac{S_{t+\Delta t} - s_t}{\Delta t} - \mathbb{E}\left[\frac{S_{t+\Delta t} - s_t}{\Delta t}\right]\right).
> $$
> The second term in this decomposition forms a martingale difference noise. Under standard Robbins-Monro step-size conditions, such noise vanishes asymptotically, ensuring that it does not affect the long-term consistency of the learning dynamics.
>
> ## On the difference between naive-dTD and dTD
> In the formulation of the loss function, the target term is detached from the computation graph (i.e., no gradient flows through it), while the prediction term is backpropagated through; please refer to Section 5.1 and Table 1 of the paper.
> As noted there, this design choice follows the standard practice in deep RL, where MSE is minimized between a fixed target and a learnable prediction.
> For the Bellman equation, the target (right-hand side of the Bellman eq.) includes the true reward $r_t$ and is generally considered more reliable than the prediction (left-hand side), which motivates treating it as fixed during training.
>
> However, the HJB equation is not merely an Itô expansion of the Bellman equation; it involves algebraic rearrangements, including swapping the left- and right-hand sides. As a result, it is not a priori clear which side is more reliable and should be treated as the target term. The choice of parameterization is therefore nontrivial. Although the naive-dTD formulation appears more conventional in structure, our proposed dTD method, which treats the value function as the trusted quantity and adjusts gradients accordingly, yields better empirical results. This behavior can be understood from the fact that the HJB equation ultimately models the evolution of the value function itself, not its gradient.
>
> ## On the direct evaluation of TD
> Value-based reinforcement learning generally follows either of two approaches: value iteration or policy iteration. As discussed in the paper, value iteration is not compatible with our formulation. In policy iteration, one alternates between policy evaluation and policy improvement. Natural extensions in this context include SARSA. While SARSA is the simplest form of policy iteration, it is incompatible with our continuous action setting. Although policy evaluation alone cannot be used as a standalone RL algorithm, it can be meaningfully incorporated into broader frameworks such as actor-critic methods. Among actor-critic variants, A2C is the most straightforward instantiation, but we chose PPO for its practicality and widespread use in deep RL today.
>
> With that being said, we fully agree with the reviewer's point that the effect of dTD would be more clearly isolated in simpler settings. For this reason, we conducted additional experiments using A2C, which are included in the table below.
>
> # Answers to Questions
> ## The assumptions of Proposition 2
> The solution to the HJB equation is known to be a viscosity solution, which is continuous but not necessarily everywhere differentiable. However, the differential operations required by the contraction analysis are applied to the function approximator, not the true value function. Since neural networks are almost everywhere twice differentiable and capable of approximating any continuous functions including the viscosity solutions. We believe this does not pose a practical issue.
>
> ## Toy Example
> We added the Pendulum task as a simple toy example (see Table A1 and A2).
>
> ## dTD with β=1
> We have added experiments for the case $\beta = 1$ to directly address this point. The results are included in Table X and support our observation that combining TD and dTD improves stability and performance across different parameter choices (see Table A1 and A2).
>
> ##  parametric representation of $V$
> All methods use the same three-layer fully connected neural network. The full implementation details are specified in the paper, and the code is available at the provided link.
>
> ## Additional Info
> We only mentioned the large discount factor regime (i.e., small discount factor in discrete time) as a possible stabilizing factor. As stated in Appendix B.2, the discount values used (0.8–0.99) are within a typical range in RL.
>
> **TableA1: Final episodic return of A2C:** Each cell shows the final episodic returns under three different noise levels: 0.00 / 0.01 / 0.05.
> |                | Pendulum         | Hopper           | HalfCheetah       | Ant               | Humanoid          |
> |----------------|------------------|------------------|-------------------|-------------------|-------------------|
> | TD             | 1000 / 1000 / 1000  | 1818 / 1846 / 1587  | 8793 / 9273 / 7904   | 1964 / 2182 / 2007   | 285 / 289 / 287   |
> | β-dTD              | 1000 / 1000 / 1000  | 2645 / 2565 / 2436  | 11948 / 10494 / 8919   | 2569 / 2469 / 2255   | 301 / 320 / 317   |
> | model-based        | 1000 / 1000 / 1000  | 2864 / 2834 / 2607   | 12182 / 12696 / 10537  | 2716 / 2892 / 2605   | 320 / 331 / 323   |
>
>
>
> **Table A2:  Final episodic return of PPO:**
> Each cell shows the final episodic returns under three different noise levels: 0.00 / 0.01 / 0.05.
>
> |         | Pendulum         | Hopper           | HalfCheetah       | Ant               | Humanoid          |
> |---------|------------------|------------------|-------------------|-------------------|-------------------|
> | model-based   | 1000 / 1000 / 1000  | 3952 / 3904 / 3707  | 20102 / 18738 / 13285   | 4202 / 4037 / 3796   | 430 / 418 / 411   |

---

> > ### Comment · Reviewer_jPHL · 2025-08-01
> >
> > A would like to thank the authors for their clarifications.
> >
> > Concerning the finite difference scheme, the clarifications you gave in your answer and indeed important and should be included in the revised manuscript.
> >
> > Concerning the difference between naive-dTD and d-TD, I now understand that it lies in the quantity which is backpropagated through or kept fixed. Despite Table 1, this was not very clear to me when I first read the paper, so I believe your explanation should be added to the main text. The discussion on the choice of parameterization is indeed very interesting, and I believe valuable for the RL and optimal control communities. Finding what is the right quantity to be parameterized (value function, policy/controller, other terms in the HJB equation) is, I believe, not a solved problem, and your experiments give preliminary hints on what should be preferred.
> >
> > Concerning the assumptions of Proposition 2, you are indeed right in pointing the fact that the parameterized version of the value function can be smooth, however what remains would be an approximation error, which is not discussed here. I don't think it is a strong limitation of the approach, but adding a small discussion on this topic would be beneficial to the paper. As mentioned in my review, I also suggest, maybe for further work, to explore the potential benefits of injecting noise into the dynamics on smoothing the true value function (hence potentially reducing the approximation error).
> >
> > Thank you also for the extra experiment on A2C, which looks convincing to me.
> >
> > Given these answers, and provided the authors incorporate those clarifications into the camera ready-paper, I update my score from weak accept (4) to accept (5).

---

### Official Review · Reviewer_q3JW · 2025-07-09

**Clarity:** 3
**Significance:** 2
**Originality:** 3
**Rating:** 5
**Confidence:** 4

**Summary:**

In the present paper, the authors introduce differential temporal difference, a model-free approach to temporal differences in reinforcement learning. To derive the method the authors build on the Hamilton-Jacobi-Bellman (HJB) equation and use Ito's stochastic calculus to derive the algorithm for differential temporal differences from the HJB as well as a number of theoretical properties such as its contraction property. This approach is subsequently evaluated on 4 continuous control environments showing it outperforming the classical temporal differences approach.

**Questions:**

### Main Questions (mostly re the evaluation)
- As mentioned above, the evaluation currently lack comparison to other algorithms to better position the work of the present paper in relation to the wider literature. In the introduction you point out a number of previous works, orthogonal to yours (line 37ff). Have you considered adding e.g. model-based approaches to TD to the evaluations to improve the paper's comparability to wider literature?
- _"It is compatible with on-policy methods such as A2C (Mnih et al., 2016) and PPO (Schulman
et al., 2017)"_ (line 44f), why is this not expanded upon with dedicated evaluations? As is, this potentially very strong evaluation claim is not backed up by the evaluation design.
- With the theoretical derivations of the new algorithm being one of the major strengths of the paper, how derived theoretical properties utilized, or even leveraged in the evaluations?

### Minor Questions
- You position the paper as mostly useful for applications in robot learning & autonomous driving, do you think that your approach could potentially also hold a lot of utility for applications in the sciences?
- Is the DEHB method used for hyperparameter tuning truly the most efficient hyperparameter tuning approach for reinforcement learning? The 2023 study referenced here as source of this knowledge is outdated. I would recommend to remove this half-sentence in light of the significant body of work on hyperparameter optimization since then. Furthermore, I would argue that their findings are no longer broadly applicable, at the same time not making the hyperparameter optimizer not all of a sudden a bad or wrong hyperparameter optimizer to choose.

**Ethical Concerns:**

["NO or VERY MINOR ethics concerns only"]

**Final Justification:**

The authors' approaches main concerns raised during review such as the limited evaluations, and the limitations of some of the theoretical results have all been addressed during rebuttal. What remains is a worthy contribution to the field, which is wholly worthy of acceptance.

**Limitations:**

Limitations only discussed implicitly throughout the paper, not explicitly in a dedicated _"Limitations"_ section. It would aid the reader considerably to summarize the limitations in one succinct limitations paragraph.

**Paper Formatting Concerns:**

- The evaluation graphs, are as is hard to read. Especially the ablations are nearly illegible. Potentially using other colors or increasing the hue factor of the standard deviation might improve the legibility considerably.

**Quality:**

2

**Strengths And Weaknesses:**

The main strengths of the paper lie in its super high clarity of the writeup throughout the paper, complementing its strong theoretical derivations. The step-by-step derivations starting out from the Hamilton-Jacobi-Bellman equation using the Ito's stochastic calculus are very clear, and easy to follow. The derived theoretical results on the contraction are a clear result of this strong theoretical foundation to the newly introduced algorithm.

The weaknesses of the paper all lie in its evaluation, to hone in on a number of the weaknesses:
- The evaluation as is show a lack of comparison to other algorithms, while utilizing standard libraries like Brax, whose interface to reinforcement learning frameworks has seen wide use. This is especially surprising given the author's acknowledgement of model-based counterparts to the proposed model-free approach of the paper, which would readily lend themselves as points of reference on the performance of the newly introduced algorithm.
- The evaluations as is, only compare to a limited number of control environments, where a larger number of (continuous) control environments are readily available through the same software packages used by the authors.
- Claims with regards to the utility of the new algorithm, such as its potential use in PPO & A2C algorithm, are not reinforced through the evaluation design.

---

> ### Author Rebuttal · Authors · 2025-07-31
>
> We thank the reviewer for the detailed and constructive feedback. We address the main concerns regarding the evaluation and clarify our design choices below.
>
> # Evaluation Updates and Justification
> ## Evaluation on A2C and Model Oracle Settings
> We originally considered PPO sufficient for our experiments, as it builds upon and improves A2C, and has been widely adopted as a de facto standard in RL research. However, we agree that the raw performance of the proposed dTD method can be more clearly assessed in A2C, where fewer confounding elements exist. We therefore conducted additional experiments using A2C, and we present only the final performance values in Table A1.
>
> We additionally addressed the concern regarding comparison with model-based methods by introducing a model oracle setting in each environment, where the drift $\mu$ and diffusion $\sigma$ terms of the environment dynamics are assumed to be known. This setup serves as a reference point for evaluating the performance of our model-free approach. The corresponding final results are also included in Tables A1 and A2.
>
> ## Comparison with Prior Work
> In our view, a valid comparison baseline must meet both of the following:
>
> ・ Estimate a Value or Q-function; \
> ・ Be compatible with deep RL frameworks.
> \end{enumerate}
>
> Among prior related studies, only Tallec et al. (2019) and Yıldız et al. (2021) satisfy the above two conditions (though they are applicable only to the deterministic setting). Although the overall approach of Tallec et al. (2019) is well-designed, its value function estimation component is essentially equivalent to classical TD, which we already included as a baseline.
> The method of Yıldız et al. (2021) relies on accurate modeling of the environment dynamics. Our additional experiment with the model oracle includes such a configuration. As we used the true drift and diffusion terms, our evaluation is free of model estimation error.
>
> As for Kim et al. (2021), their method focuses on improving the computation of the $\arg\max_a Q(s, a)$ operation, whereas our method modifies the TD update itself. It means that they target fundamentally different components of the RL algorithm. Comparing them is like comparing apples and oranges --- these are orthogonal contributions. In fact, our method could in principle be combined with HJ DQN, for example, by incorporating our TD update into Equation (22) of Kim et al. (2021).
>
> Other related studies mentioned in the paper are rooted in stochastic control theory, typically under limited settings such as finite-horizon or LQR formulations. These settings often require additional assumptions, such as time-dependent value functions $V(t, s_t)$ in the finite-horizon case, and are incompatible with modern deep RL pipelines and standard benchmark environments.
>
> ## Number and Scope of Evaluation Environments
> The aim of the work is to introduce a theoretically grounded model-free loss function for continuous RL and to show that HJB-based TD methods, previously considered incompatible with deep RL, can actually be implemented and run within standard deep RL pipelines.
> Rather than focusing on achieving state-of-the-art results, our primary contribution is a conceptual one, particularly towards bridging stochastic optimal control and model-free RL.
> We not only formulated the loss function but also demonstrated that it works with commonly known benchmarking tasks.
> We therefore respectfully disagree with the concern regarding the number of evaluation environments.
>
> Our experiments are on four standard continuous control tasks, spanning a wide range of dimensionality from 10 to 244. We believe these environments are representative and sufficient to demonstrate the behavior of our method. The other gym tasks such as Pusher or Walker2d are unlikely to yield fundamentally new insights compared to the environments already included.
> With that being said, we do acknowledge the value of including simple, low-dimensional environments to complement our analysis and provide additional clarity. To that end, we have added experiments on the Pendulum environment and included the results in Tables A1 and A2.
>
> # Answers to Remaining Questions
> ## Empirical Reflection of Theoretical Insights
> The contraction property derived in the theory implies that the convergence of our method is faster when the environment dynamics are smoother (i.e., have a smaller Lipschitz constant). Since our method targets the same objective function as classical TD, both approaches are expected to converge to similar value functions asymptotically. This theoretical insight is empirically reflected in the experimental results and discussed in the discussion section.
>
> ## Applicability to Scientific Domains
> Yes. In fact, recent work in diffusion models has applied stochastic control to enhance the aesthetic quality and prevent distorted contents [R1,R2]. These approaches typically require estimating the drift term of the reverse diffusion process, which in turn necessitates training an additional model solely for this purpose. Our method offers a theoretically grounded and model-free way to avoid this drift estimation step and may offer significant advantages in such applications.
>
>
> ## Revised Wording Regarding DEHB
> We agree with the reviewer. The current phrasing will be replaced with a more neutral statement to avoid overstatement, e.g.: "We applied DEHB, a multifidelity optimization method that showed strong performance in prior work."
>
> We hope that the additional evaluations and clarifications provided above help address the concerns.
>
>
> **TableA1: Final episodic return of A2C:** Each cell shows the final episodic returns under three different noise levels: 0.00 / 0.01 / 0.05.
> |                | Pendulum         | Hopper           | HalfCheetah       | Ant               | Humanoid          |
> |----------------|------------------|------------------|-------------------|-------------------|-------------------|
> | TD             | 1000 / 1000 / 1000  | 1818 / 1846 / 1587  | 8793 / 9273 / 7904   | 1964 / 2182 / 2007   | 285 / 289 / 287   |
> | β-dTD              | 1000 / 1000 / 1000  | 2645 / 2565 / 2436  | 11948 / 10494 / 8919   | 2569 / 2469 / 2255   | 301 / 320 / 317   |
> | model-based        | 1000 / 1000 / 1000  | 2864 / 2834 / 2607   | 12182 / 12696 / 10537  | 2716 / 2892 / 2605   | 320 / 331 / 323   |
>
>
>
> **Table A2:  Final episodic return of PPO:**
> Each cell shows the final episodic returns under three different noise levels: 0.00 / 0.01 / 0.05.
>
> |         | Pendulum         | Hopper           | HalfCheetah       | Ant               | Humanoid          |
> |---------|------------------|------------------|-------------------|-------------------|-------------------|
> | model-based   | 1000 / 1000 / 1000  | 3952 / 3904 / 3707  | 20102 / 18738 / 13285   | 4202 / 4037 / 3796   | 430 / 418 / 411   |
>
>
> [R1] Masatoshi Uehara et al.(2024) Understanding Reinforcement Learning-Based Fine-Tuning of Diffusion Models: A Tutorial and Review. arXiv:2407.13734, 2024.
>
> [R2] Wenpin Tang.(2024) Fine-tuning of diffusion models via stochastic control: entropy regularization and beyond. arXiv:2403.06279, 2024.

---

> > ### Comment · Reviewer_q3JW · 2025-08-05
> > **Thank you for the Clarification**
> >
> > Thank you so much for the detailed response, to lead off the newly arisen questions, are the results reported in the final table the ones of a moving average window, or are they just the last episodic return. Potentially taken as the mean?

---

> ### Author Response · Authors · 2025-08-06
> **Clarification on Final Return Reporting**
>
> Thank you for your question. The reported results are the mean final return over one episode, averaged across multiple runs, after $2.5\times10^6$ training steps. As a supplementary note, for the pendulum toy problem, while the final return reaches the maximum of 1000, the learning curves show that dTD converges more quickly, so this experiment still yields meaningful results.

---

> > ### Comment · Reviewer_q3JW · 2025-08-07
> > **Thank you for all the Clarifications**
> >
> > I want to thank the authors for their extensive clarifications, most of my concerns have been addressed during rebuttal and as such I've increased my rating.

---

### Decision · Program_Chairs · 2025-09-17

**Decision:**

Accept (poster)

**Comment:**

This paper introduces differential TD learning, derived from the HJB equation. Reviewers appreciated the rigor of the derivations and the potential to bridge stochastic control with RL. The main weakness is the limited empirical validation, with missing baselines and restricted scope of experiments. Still, the methodological contribution is solid and judged valuable for the community. The decision is accept.